# Learned feature representations are biased by complexity, learning order, position, and more

**Andrew Kyle Lampinen**                                          *lampinen@google.com*
*Google DeepMind*

**Stephanie C. Y. Chan**                                          *scychan@google.com*
*Google DeepMind*

**Katherine Hermann**                                             *hermannk@google.com*
*Google DeepMind*

**Reviewed on OpenReview:** *https://openreview.net/forum?id=aY2nsgE97a*

## Abstract

Representation learning, and interpreting learned representations, are key areas of focus in machine learning and neuroscience. Both fields generally use representations as a means to understand or improve a system's computations. In this work, however, we explore surprising dissociations between representation and computation that may pose challenges for such efforts. We create datasets in which we attempt to match the computational role that different features play, while manipulating other properties of the features or the data. We train various deep learning architectures to compute these multiple abstract features about their inputs. We find that their learned feature representations are systematically biased towards representing some features more strongly than others, depending upon extraneous properties such as feature complexity, the order in which features are learned, and the distribution of features over the inputs. For example, features that are simpler to compute or learned first tend to be represented more strongly and densely than features that are more complex or learned later, even if all features are learned equally well. We also explore how these biases are affected by architectures, optimizers, and training regimes (e.g., in transformers, features decoded earlier in the output sequence also tend to be represented more strongly). Our results help to characterize the inductive biases of gradient-based representation learning. We then illustrate the downstream effects of these biases on various commonly-used methods for analyzing or intervening on representations. These results highlight a key challenge for interpretability—or for comparing the representations of models and brains—disentangling extraneous biases from the computationally important aspects of a system's internal representations.

## 1 Introduction

A key goal of machine learning research is learning representations that allow effectively performing downstream tasks (Bengio et al., 2013). A growing subfield aims to interpret the mechanistic role that these representations play in the models behaviors (Geiger et al., 2021; Olsson et al., 2022; Geiger et al., 2024; Nanda et al., 2023; Conmy et al., 2023; Merullo et al., 2023; Wu et al., 2024), or to alter those representations to improve model alignment, interpretability, or generalization (Sucholutsky et al., 2023; Bricken et al., 2023a; Zou et al., 2023). Likewise, neuroscience studies the neural representations that a system develops and how they relate to its behavior (Churchland & Sejnowski, 1990; Baker et al., 2022). Each field focuses on representations as a means to understand or improve a system's computations—that is, its more abstract patterns of behavior on a task, and how those behaviors are implemented.

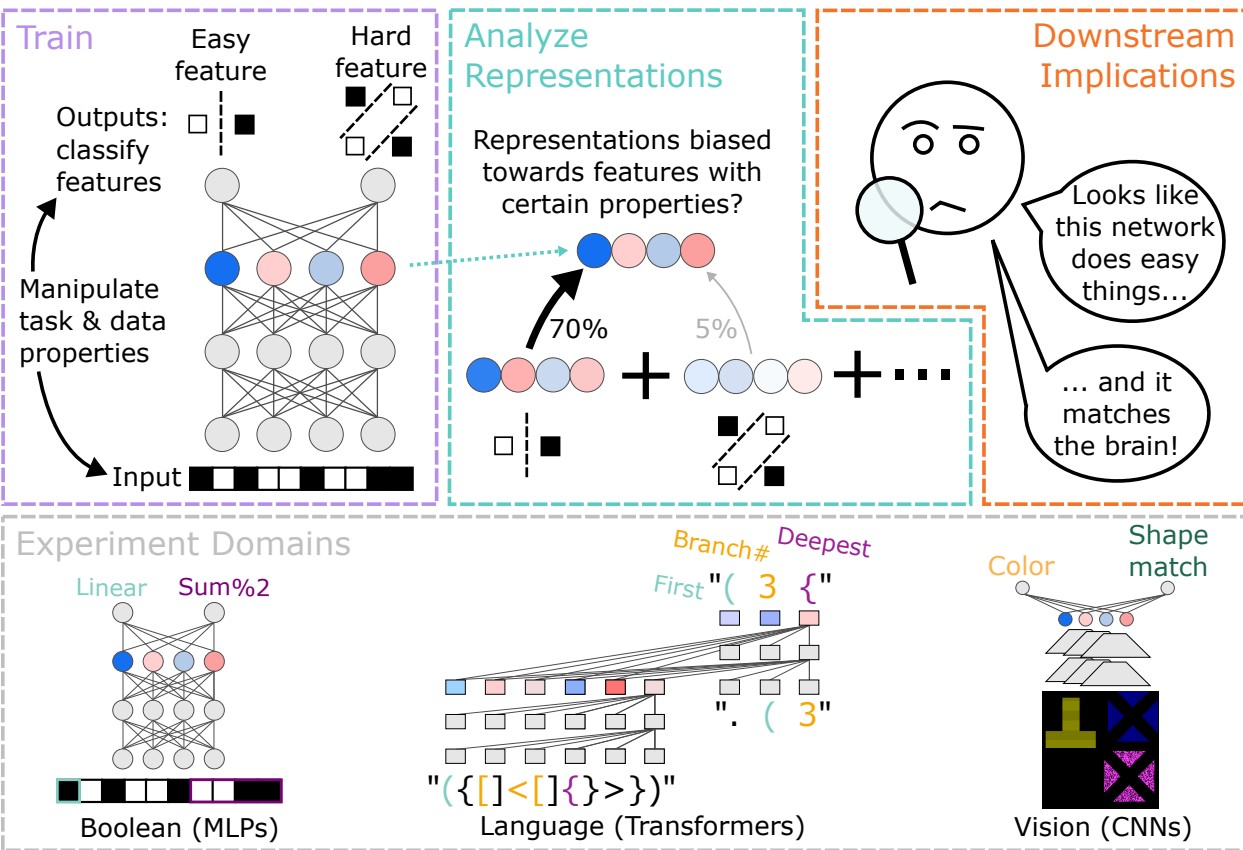

Figure 1: An overview of our approach. (top left) We train networks to compute multiple features of an input, on controlled datasets where we systematically manipulate various properties of the input and target output distribution; for example, to compute easier- and harder-to-compute features of the input. (top center) We study how these properties bias the feature representations the model learns—for example, towards representing easier-to-compute features more strongly. (top right) We explore the impacts of those biases on downstream areas like interpretability and cognitive neuroscience. (bottom) We consider these phenomena across a range of experiment domains, including boolean functions and various language and vision tasks, using standard architectures for each domain.

However, the relationship between representation and computation is nontrivial. Thus, using representations to understand or improve computation depends fundamentally on the *linking assumptions* we make about their relationship. Here, we attempt to shed some empirical light on this linking, by characterizing how computational role, and various aspects of the features and extraneous details of the learning process, affect the representations that deep neural networks learn.

To do so, we train various models—MLPs, ResNets and Transformers—to compute *multiple* independent abstract features about an input, through separate outputs (Fig. 1). This approach provides a controlled setting in which each abstract input feature plays an equivalent computational role: contributing to predicting exactly one output value. In this controlled setting, we manipulate various properties like feature complexity, the order in which features are trained, and their prevalence and distribution within the inputs. We show that, although these properties do not substantially affect the computational role that each feature plays in the model's input-output behavior, they substantially bias the structure of the model's learned representations.

In particular, we find that:

- Features that are simpler to compute (e.g. linearly computable from the input) tend to be more strongly represented by the model than complex (e.g. non-linear) features that play an equivalent computational role in the model's outputs.

- Features that are learned earlier tend to be represented more strongly than those that are learned later. Thus, learned representations are substantially path-dependent (cf. Lubana et al., 2023)—if a model is pretrained to compute feature A, then trained to additionally compute feature B, it will have different representations than if it was trained on feature B before feature A, or if it was trained on both A and B simultaneously—even with similar behavior on train and test sets.

- Other properties like the prevalence of a feature in the training dataset, or the output position in which the model is trained to report it, can likewise bias feature representations.

- These broad patterns of biases are common across a range of models and optimizers; they seem to be rather general patterns of gradient-based deep representation learning rather than artifacts of a particular architecture. However, we note there are complex interactions between architecture, optimizer, and task, which can shift or even reverse the biases in some cases.

- These feature representation biases have downstream implications for interpreting learned representations, comparing them between different systems, or trying to alter them. Conclusions or results may be biased by relatively extraneous details of the learning process.

In Sections 7 & 8 we set these results in the broader context of related work, such as prior work on the computational- and feature-level inductive biases of deep learning, and challenges of interpretability. Our results may have implications for both artificial intelligence and cognitive (neuro)science.

## 2 Definitions & methods

Our goal is to study how feature representations may differ, even when the features play similar computational roles. But what is a feature or a representation? Many of the terms we use are heavily overloaded, or even philosophically disputed. Thus, we begin by defining the way we will use them here.

**Input stimulus:** a raw input $I$ provided to a model, for example a tokenized string, a vector of boolean values, or an image.

**Feature:** an abstract, task-relevant property of an input; e.g., whether "to be or not to be" appears within a sentence, or whether XOR of the first and third boolean value is true. Thus, a feature can be defined as the result of applying some function to the input: feature value $= f(I)$. Generally, the features we discuss are those a model is explicitly trained to output.

**Computation:** the behavior (i.e. input-output mapping) of a system, described at the abstract level of the task-relevant features. For example, if we train a vision model to report the shape and color of an object in an image, and it does so even on a held-out test set of images, we say that the model is computing the features shape and color. That is, the task of the model would be to output the vector $[f_{shape}(I), f_{color}(I)]$ where the feature functions identify the respective values.

**Computational role of a feature:** the downstream contribution of a feature to the desired output behavior, once that feature's value has been computed. For example, suppose the network is trained to label sentences with whether they contain the word "cake", and whether they are grammatical, through separate outputs ($[f_{cake}(I), f_{grammatical}(I)]$). We say these features play an equivalent computational role (direct output) once they are computed, even though computing the value of one feature might be much easier than the other.

**Representation:** a distributed activity pattern across neural units within (part of) a model. We will denote the representation the model produces for an input $I$ as $\phi(I)$.

**Feature Representation:** the portion of a representation (e.g. a vector subspace) that can be predicted from knowing the value of a particular feature for the current input. (Note that we do *not* here stipulate that this subspace must be causally implicated in the model's behavior, unlike, e.g., Cao, 2022. We make this choice for practical reasons, to avoid verifying causality throughout training. However, we do verify the causal role of the representations in some cases in §6.1.)

**Representation variance explained by a feature:** the proportion of the total variance across representations produced by a set of stimuli that is accounted for solely by knowing that feature's value for each stimulus; i.e., the $R^2$ of a linear regression from the feature values for each stimulus onto the representations produced for those input stimuli. See Methods below for a more formal definition. This is a method for quantifying the "strength" of a feature in the representation.

In section 6 we justify some of these choices empirically, for example showing that the representations are causally implicated in each feature's input-output mapping, illustrating how the representation variance explained by a feature impacts interpretability methods and RSA, and examining how feature variance in the representations affects downstream learning.

## 2.1 Methods

In Fig. 1 we show an overview of our approach. We primarily train networks to output classifications of multiple features, through separate output units (e.g. from an MLP), or as a sequence (from a Transformer). We construct the datasets such that the features are statistically independent from one another. In this setting, we know the "correct" computation that the model should learn. We train models on large enough datasets, and for long enough, that they achieve high accuracy (>95%, and often 100%) on the held out test set for all features (except where explicitly noted in the text). Thus, we can ensure that the models are computing each feature in a way that broadly generalizes in accordance with our understanding of that feature.

We investigate whether particular properties of the features change the way that those features are represented. These properties included feature complexity, feature prevalence in the training dataset, or feature position in the output sequence. We therefore create families of training datasets which systematically manipulate these properties. We also create held-out validation and test datasets corresponding to each training dataset, for analysis and to ensure that the model generalizes as expected. The train/validation/test datasets are sampled IID.

To analyze the results, we extract internal representations from the trained model for each stimulus in the validation or test set, and then inspect those representations are driven by each feature. Our main analyses focus on the variance explained for each feature. To compute this quantity, we fit linear regressions whose inputs are the value of a feature, and whose outputs are the activations of each unit in the model's representations. We measure the variance explained by a feature as the $R^2$ resulting from the corresponding regression, when the regression is fit using the validation set and tested on the test set.

More formally, if $\phi(I_i) \in \mathbb{R}^n$ denotes the model's learned representation vector for an input $I_i$,[1] and $f(I_i)$ is the corresponding feature value, then via linear regression we find the optimal linear predictor $W_f^*$ of the model's representations on the validation set $V$ from that feature:

$$W_f^* = \min_W \mathbb{E}_{I_i \in T} \left[ ||W f(I_i) - \phi(I_i)||^2 \right]$$

We then assess the variance explained by that feature relative to the total variance in the model's representations on the test set $T$:

$$R_f^2 = \mathbb{E}_{I_i \in T} \left[ 1 - \frac{||W_f^* f(I_i) - \phi(I_i)||^2}{||\phi(I_i)||^2} \right]$$

where the total variance $\mathbb{E}_{I_i \in T} ||\phi(I_i)||^2$ is simply the variance of the representation units across all inputs in the dataset.[2] When comparing representation variance explained over the course of training, we normalize each timepoint relative to the total variance at the *end* of training (i.e., in the equation above, we replace the total variance term $||\phi(I_i)||^2$ in the denominator with the final value $||\phi_{final}(I_i)||^2$ at all timepoints), to plot all values on a consistent scale.

---

[1] With the mean representation across the dataset subtracted out, for notational convenience when writing the variance.

[2] Note that when we compare multiple features, this measure could in principle "overcount" variance by attributing the same variance in the representations to both features. However, because we have constructed the datasets such that the features are statistically independent, we do not have this issue in practice.

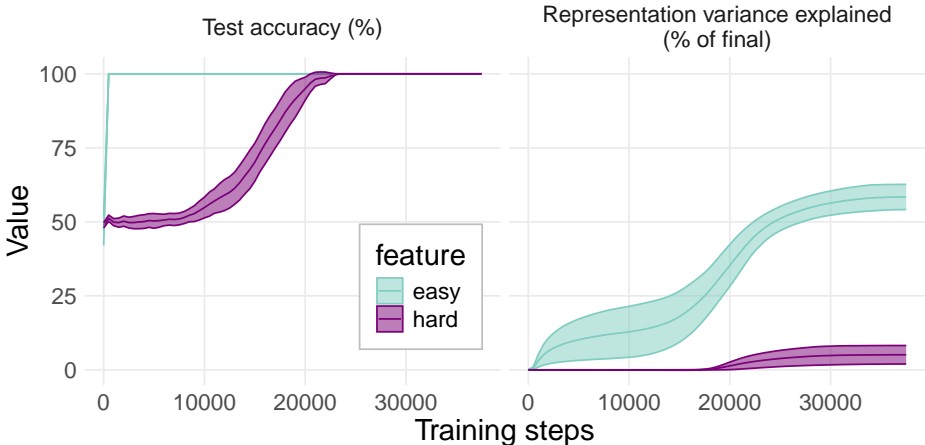

Figure 2: Test accuracy (left) and representation variance (right) over learning in an MLP trained to output easy and hard features. The easy feature (reading out a single input) is learned quite rapidly, but the hard feature (sum of 4 inputs mod 2) is learned more slowly; nevertheless, by the end of training the model achieves perfect accuracy for both features on a held-out test set. Despite this, the easy feature occupies substantially more of the penultimate layer representation variance than the hard feature. It is particularly interesting to note that long after the easy feature has been mastered, the representation variance attributable to it continues to grow, *especially* when the harder feature performance is most rapidly improving. (Representation variance is plotted normalized by the total representation variance at the end of training, to show the gradual increase over time. See Appendix 20 for an unnormalized version. Lines are averages across 10 seeds, intervals are 95%-CIs.)

## 3 MLPs on binary features

We begin with some simple experiments on MLPs trained to compute binary features from vectors of 32 binary inputs. This initial setting is motivated by the original results of Hermann & Lampinen (2020) on RSA, and provides a simple domain where the complexity of relationships between input features and their outputs is clearly defined.

### 3.1 An initial look at complexity bias

We first train a 4-layer MLP to simultaneously output the values of two features embedded amongst other noisy inputs: (1) an easy, linear feature (reading out the value of an input unit) (2) a harder feature (the sum mod 2 of a non-overlapping set of 4 inputs).[3] That is, suppose the input vector $I$ is composed of $k$ boolean inputs $I = [A, W, X, Y, Z, N_1, ... N_{k-5}] \in \{0, 1\}^k$, where the first 5 will be relevant to one of the features and the remaining $N_1 ... N_{k-5}$ are purely noise that the model must learn to ignore. Then the features are defined respectively as:

$$f_{\text{Easy}}(I) = A$$
$$f_{\text{Hard}}(I) = (W + X + Y + Z) \mod 2$$

We study the representations of these two features at the penultimate layer of the model, immediately before the output logits.

In Fig. 2 we show the basic pattern of results, which illustrates many of the phenomena that we study below. In the left panel, we plot model accuracy on a held-out test set over training for each feature (see Appendix B.1 for the corresponding loss curves). The easy linear feature is learned quite rapidly, almost instantly on

---

[3]The nonlinear feature is harder by a variety of overlapping measures, including the minimum depth and width ReLU network that can compute the feature, the number of updates required to learn it, the dataset size required to learn this feature in a way that generalizes, etc.

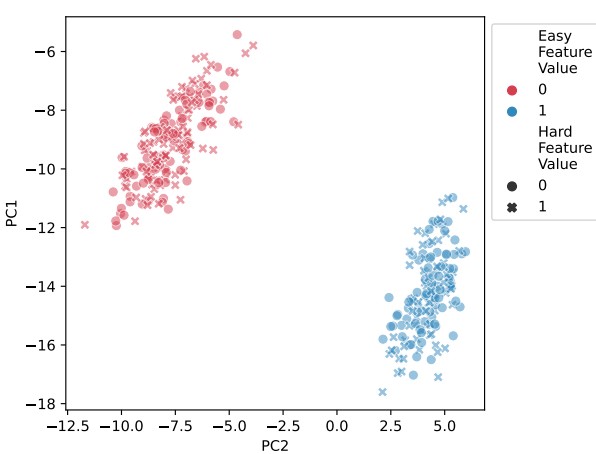

Figure 3: The first two principal components of the penultimate representations of an MLP after learning the easy (linear) and hard (sum % 2) features (see Fig. 2). The first PCs of the representations are structured into two clusters defined entirely by the value of the easy feature (colors); the hard feature (shapes) does not substantially impact the top PCs.

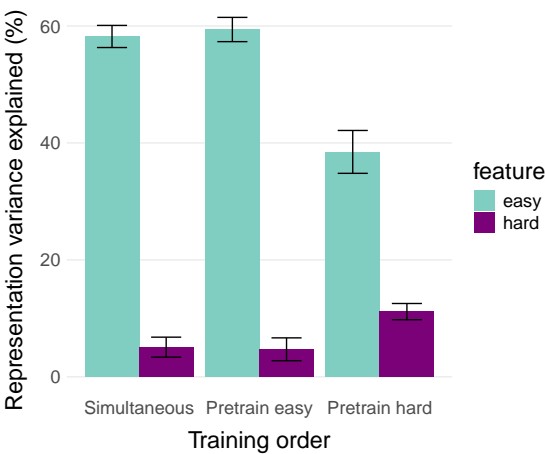

Figure 4: Penultimate representation variance explained by the easy (linear) and hard (sum of 4 inputs mod 2) features, across different training paradigms. Pretraining to output the easy feature results in very similar representations to training the features simultaneously, since the easy feature is learned first anyway. Pretraining the hard feature closes the gap somewhat, but the easy feature still dominates. (Bars are averages across 10 seeds, intervals are 95%-CIs.)

the scale of the plot.[4] By contrast, the hard feature is learned more slowly. In the right hand panel, we plot the portion of the variance in the penultimate representations of the model explained by each feature. Of course, because the easy feature is learned early in training, it initially begins to occupy more variance in the model feature representations. As the harder feature begins to be learned, it too occupies some variance in the penultimate representations. However, the easy feature dominates the hard one; it occupies far more variance. In fact, even though the easy feature is learned and generalized extremely early in training, as the hard feature begins to be learned, there appears to be some *cooperative* pressure that actually *inflates* the total variance occupied by the easy feature far beyond its initial value (note the inflection point in the easy feature variance explained). Thus, throughout training, the penultimate representations of the model are driven mostly by the easy feature, and learning the hard feature actually increases this dominance.[5]

To visualize this result more intuitively, in Fig. 3 we show the representations at the end of training for one of the models, projected down to the top two principal components (PCs). The representations are entirely structured into two clusters defined by the value of the easy feature; there is little impact of the harder feature. Of course, because the hard feature *is* represented by the model in a generalizable way, it *must* be encoded in the representations somewhere, and indeed in §6.2.1 we show that it is captured in higher principal components, depending on which other features are being learned. However, simply from looking at the reduced-dimensionality representations shown here, it might seem that the model is doing something quite simple, focused only on the value of the easy feature.

## 3.2 Is the effect due to feature complexity or simply learning order?

Where does the gap in representational variance between the easy and complex features originate? Is it due to the effect of feature complexity in itself, or the fact that the easy feature is learned earlier? To disentangle

---

[4]In fact, the easy linear feature is linearly decodable with $\geq 98\%$ test accuracy from the penultimate representations of an *untrained* network.

[5]We provide a simple colab demonstrating this result at `https://gist.github.com/lampinen-dm/b6541019ef4cf2988669ab44aa82460b`

the contributions of these two factors, in this section we experimentally manipulate the training order—either training both features simultaneously, as above, or pretraining one feature to convergence before beginning to train the other. By pretraining the more difficult feature, we ensure that it is learned first, and disentangle the effect of learning order from the effect of difficulty.

In Fig. 4 we show the results. Pretraining the easy feature does not substantially change the results, which is sensible since the easy feature is learned first regardless. Pretraining the hard feature does close the gap somewhat, but the easy feature still dominates the representations. See Appx. B.3 for learning curves.

### 3.3 Varying features and accounting for many ways they can be represented

If learning order alone does not explain the dominance of the easy feature, what aspects of feature complexity might? To explore this, we trained models on a larger set of hard features, ranging in difficulty from simple AND of groups of inputs to more complex ones. In analyzing these models we considered another property of nonlinear features: there are more reasonable ways to compute them. For example, XOR(X,Y) can be decomposed as AND(OR(X,Y), NOT AND(X,Y)) or as OR(AND(X, NOT Y), AND(NOT X, Y)), which yield different patterns of representations (cf. Fig. A.14 in Hermann & Lampinen, 2020). By contrast, all simple ways of encoding the linear feature are linearly equivalent. Could it be that the model is strongly representing *components* of the hard feature, even if it is not representing the hard feature itself?

To test this possibility, we measured the variance explained by *all* patterns of inputs relevant to the hard feature. That is, we enumerated all the possible hard-feature-relevant input patterns and then created a one-hot embedding for each of them. We then regressed from this set of labels onto the representations. The advantage to this analysis is that *any* function of the hard inputs alone is linear in this input-pattern space—it is essentially a tabular function representation where there is a separate entry for each possible input pattern—and thus it can account for any linear and non-linear patterns of encoding the hard input, as long as they do not mix in other irrelevant inputs. (Note that the easy feature inherently captures all relevant input patterns, so there is no need to perform an equivalent analysis for the easy feature.)

The results of this analysis are shown in Fig. 5. In sum, accounting for the variance of all input patterns does not by itself explain the gap between easy and hard features; if the features are trained simultaneously the gap remains quite large. However, if the hard feature is pretrained, *and* the analyses are performed with respect to all hard input patterns, the gap vanishes, and there is even a slight bias towards the hard feature in some cases. Thus, the easy-hard gap seems to be explained by a combination of hard features being learned later, and the fact that there are more ways to represent harder features.

### 3.4 Additional evaluations of complexity biases on unit level, and with additional features

We present some additional evaluations in full detail in the supplement, but summarize them here.

**Per-unit analyses:** We find similar qualitative biases, as well as interesting individual effects, when analyzing each unit in the penultimate layer individually (Appendix B.4).

**Sparsity:** We also find representation sparsity biases: more complex features tend to produce sparser representations (Appendix 24).

**More than two features:** Finally, in Appendix B.6 we show that there are qualitatively similar, graded biases when the models are trained simultaneously on more features sampled from a set of varying difficulty.

### 3.5 Feature prevalence also biases representations

In the above experiments, all features were balanced to appear equally often in the dataset. However, distributions of properties in naturalistic data tend to be skewed (e.g. Piantadosi, 2014), and these distributional properties can shape the solutions that models learn (e.g. Chan et al., 2022). Thus, we explored whether feature prevalence affects the representational biases we observed above.

To do so, we constructed datasets containing two linear features (easy), and two sum-mod-2 features (hard), where the first feature of each type was present (i.e. had a label of 1) in 50% of the data (common), but

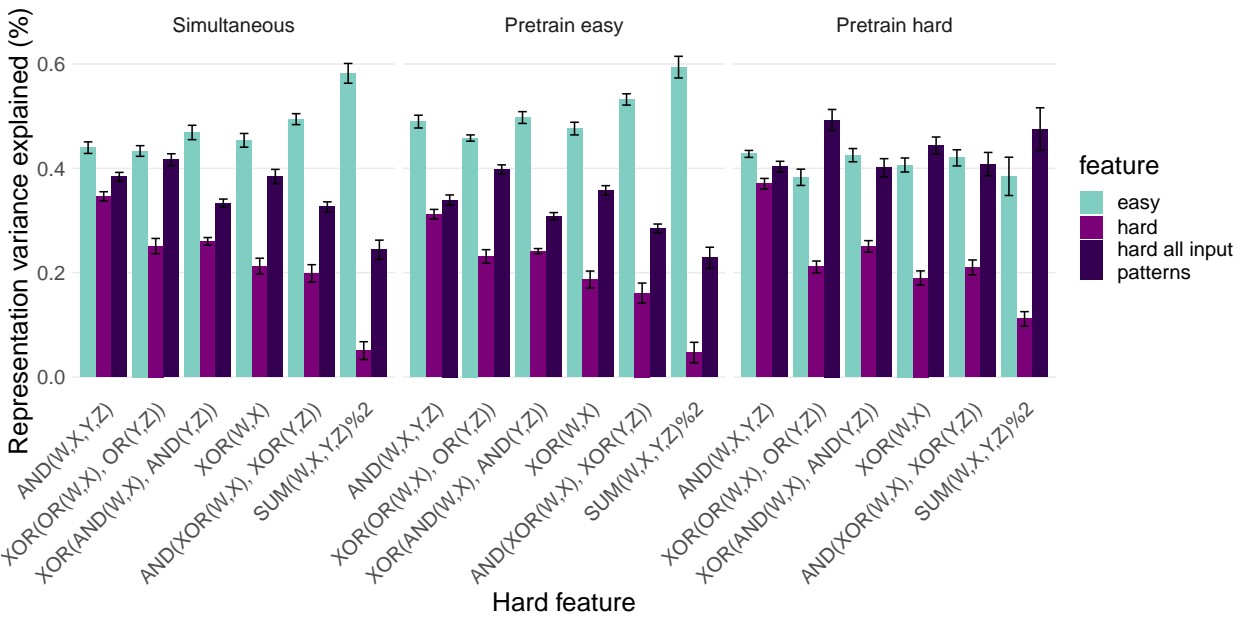

Figure 5: Penultimate representation variance explained at the end of training by the easy (linear) and hard features, as well as all the patterns over the inputs relevant to the hard feature, and across different training paradigms.

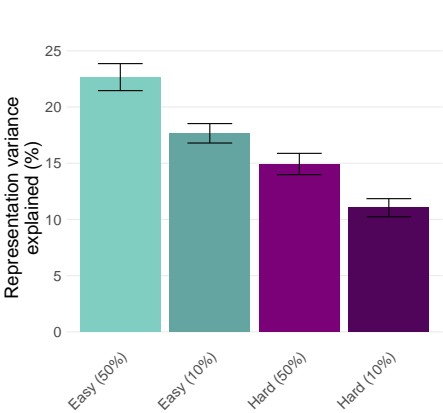

Figure 6: Feature prevalence, as well as difficulty, affects representational variance in MLPs. Features that appear less frequently in the training data tend to explain less variance in the model's penultimate representations than more prevalent features of equal difficulty. (Bars are averages across 32 seeds, intervals are 95%-CIs.)

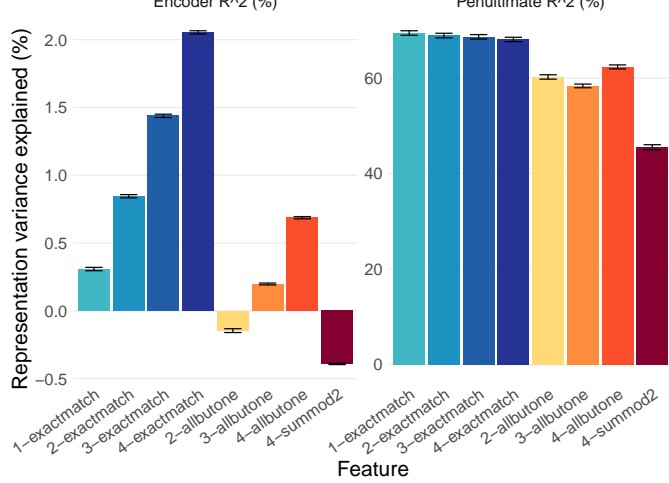

Figure 7: Representation variance explained at the final encoder layer (left), and pre-logits decoder layer (right) for transformers trained on character sequence classification datasets. In the encoder, all features explain relatively little of the overall variance (note vertical axis scale). There are strong effects of number of tokens involved, as well as complexity. In the penultimate decoder layer (right), the features follow the expected complexity ordering, though there appears to be some effect of number of tokens. (Bars are averages across 128 seeds, intervals are 95%-CIs.)

the other appeared in only 10% of the data (rare). For example, for the easy rare feature, the corresponding input would be 0 in 90% of the dataset examples. We analyzed how difficulty and prevalence jointly influence representations.

In Fig. 6 we show the results. Feature prevalence (in addition to difficulty) indeed affects the feature representations. This is somewhat intuitive, as more frequently occurring features effectively receive more gradient updates, and thus are learned earlier. However, again the model achieves perfect accuracy and very low loss across all features; thus, it would not be immediately obvious from the model's behavior that one feature should be represented more strongly than another because of a frequency artifact in the training distribution. This result provides yet another demonstration that when a model that is computing effectively the same thing—generalizing to the test set for both the frequent and rare version of each feature—it can nevertheless implement those computations in ways that are biased towards one feature or another.

### 3.6 The effects of architecture and hyperparameters—generally weak but complex interactions

How general are our findings? In Appx. B.7 we explore the effects of MLP architecture and other hyperparameters. We summarize the results here. First, deeper MLPs tend to show larger representational gaps between easy and hard features; model width has little effect. Despite the fact that nonlinearities strongly influence the structure of learned representations (Saxe et al., 2019; Alleman et al., 2024), our general pattern of findings is robust to choice of nonlinearity (tanh or rectifier).

**Optimizers and dropout:** There is, however, a surprising interaction between dropout and choice of optimizer at convergence. Without dropout, all optimizers show similar effects, with large gaps between easy and hard features. However, we show in Appx. B.7.1 that dropout yields strikingly different effects with different optimizers at convergence; while SGD and AdaGrad show consistent gaps between easy and hard features, other common optimizers like Adam show qualitatively different effects with dropout, and in fact show a slight hard feature bias at moderate dropout rates. However, this is only true at convergence— earlier in training, even when the hard features have been learned and generalize quite well, the easy features remain dominant. Furthermore, we also show that transformers trained on more complex datasets (see below) show more consistent feature representation biases across dropout levels.

## 4 Transformers on sequence tasks

The majority of models employed in modern deep learning are not MLPs. In recent years, Transformer (Vaswani et al., 2017) models have come to dominate, in language and beyond. We therefore assessed whether transformers show similar effects to those we observed in simpler architectures, and whether the additional properties of language-like data—for example, positions and structure of tokens within a sequence—likewise bias feature representations.

To do so, we instantiated several increasingly language-like training regimes. The simplest are character-level sequence-to-sequence language datasets that bear some similarity to those used above, but with a few additional properties. These datasets involve mapping a sequence of letters to a sequence of classifications of non-overlapping subsequences of the original sequence. In particular, we consider three types of features. The simplest is exact-match vs. non-match to a particular letter sequence (analogous to the linear feature above), for example 1 if the sequence is 'A,B,C' and 0 otherwise. We also tested two increasingly nonlinear features: matching all-but-one of the letters in the sequence (XOR-like), and the sum of matches modulo 2 (analogous to the most complex feature above). In addition, we evaluate the effect of the number of letters in the target sequence, and the position within the output sequence. Specifically, we created datasets containing 8 features (exact match of 1-4 token sequences, all-but-one for 2-4 token subsequences, and sum-mod-2 for 4 token subsequences), along with distractor characters in between. We sampled the feature order randomly for each dataset.

We trained encoder-decoder transformers on 128 of these datasets and evaluated the representation variance explained in the last representation of the encoders (after concatenating all token representations), and the variance explained immediately before the logit layer of the decoder *when outputting that feature*. That is, unlike in the MLP experiments above, in the decoder the representations of different features are no longer

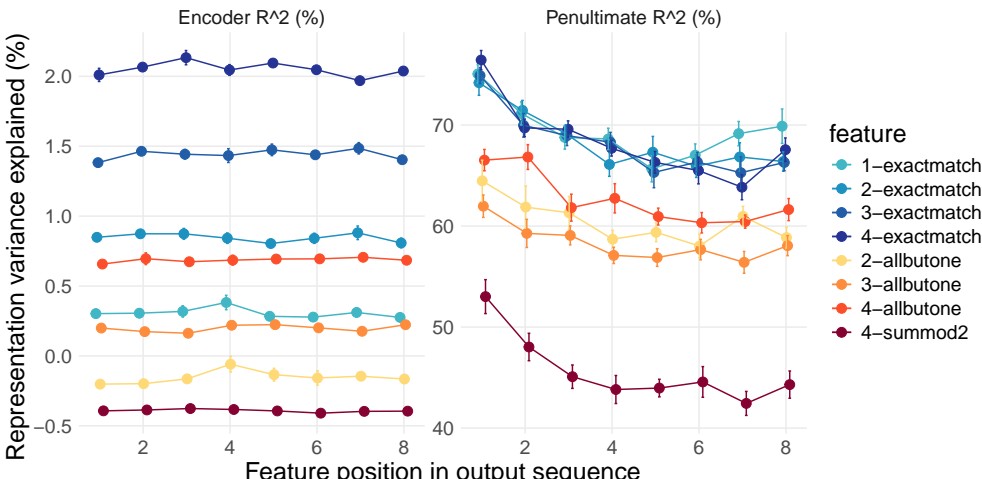

Figure 8: Position in the output sequence affects representation variance explained at the pre-logits decoder layer (right), but has little effect at the final encoder layer (left), for transformers trained on character sequence classification datasets. Features that appear earlier in the sequence are generally represented more strongly. (Lines are averages across 128 feature orderings, errorbars are 95%-CIs.)

being analyzed simultaneously on the same representations, instead they are analyzed at different positions in the output sequence.

In Fig. 7 we show the results. There are several interesting phenomena to note. First, none of the features explains very much of the overall variance in the penultimate encoder representations. (Note that the variance explained can be negative due to failure to generalize, because we fit the regressions using a validation set and compute variance explained on the test set.) Nevertheless, there are clear biases based on both complexity and the number of tokens involved in computing the feature; generally features involving more tokens are represented more strongly, while more complex features are represented less strongly.

In the penultimate decoder layer, we observe results that are qualitatively similar to those observed in MLPs; simpler features explain more variance than more complex ones. Number of tokens does still have some effect, but the pattern is less clear. Moreover, we observe substantial output order effects on representations in the decoder (Fig. 8)—in general, features earlier in the sequence occupy more variance.

**Sequential training:** sequential training of one feature at a time also yields interesting biases, which shift over training in distinct ways for the encoder and decoder (Appx. B.8).

### 4.1 Dyck languages: structural and token-level properties dominate in different model components

We next trained transformers on a set of more complex tasks involving processing hierarchical structures from string inputs. More specifically, these tasks are based on the Dyck languages—strings of balanced brackets. These languages have been used in a variety of prior works studying generalization and interpretability in RNNs (Hewitt et al., 2020) and Transformers (Yao et al., 2021; Murty et al., 2023; Friedman et al., 2023a). Like several prior works, we create tasks based on the versions of these languages with bounded depth (Hewitt et al., 2020). We sample strings from Dyck-(20,10); that is, balanced strings over 20 brackets with nesting depth at most 10.

However, in contrast to prior works (which have mostly focused on language modeling), we train transformers to output a variety of classifications of these sentences, including structure-independent token-level "semantic" features (a binary classification of the first bracket type), analogous token-level but structure-dependent features (classifying the type of the first bracket at maximum depth), or purely structural features (the maximum depth, number of root brackets, and maximum branching factor) of the sequence. These features do not capture the full structure of the sentences, but allow for more controlled experiments and analyses than

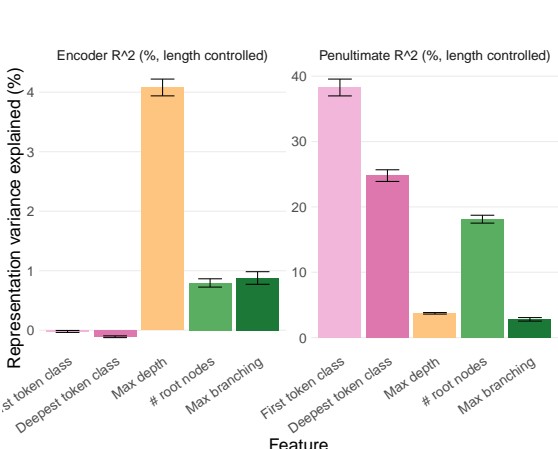

Figure 9: Representation variance explained at the final encoder layer (left), and pre-logits decoder layer (right) for transformers trained to identify semantic and structural features of sentences from Dyck-(20,10). Again, there is a bias towards simpler features at the penultimate layer; at the encoder layer, however, the most variance is explained by depth. (Because sequence length correlates with features, this plot controls for sequence length; raw results are reported in Fig. 35. Bars are averages across 32 seeds, intervals are 95%-CIs.)

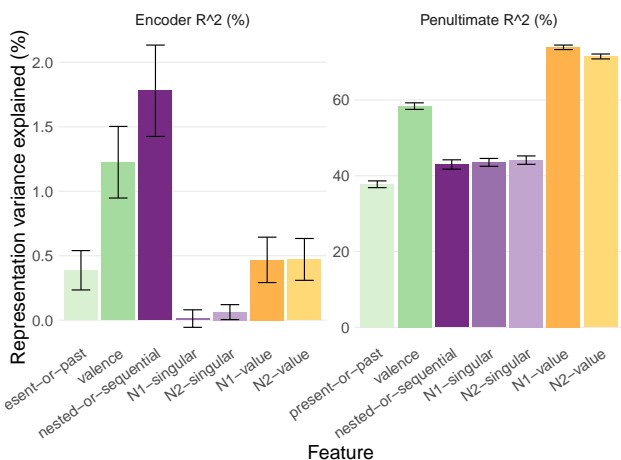

Figure 10: Representation variance explained at the final encoder layer (left), and pre-logits decoder layer (right) for transformers trained to identify semantic and syntactic features of sentences sampled from a semi-naturalistic grammar. In the encoder, structural syntactic features dominate, but there is notable variability among the other features. At the output, there are strong effects of number of possible output values for a feature—features like identifying the noun that plays a particular role within the sentence (yellow) are much more strongly represented than abstractions about that noun that only take two values (such as whether it is plural or singular, light purple). (Bars are averages across 32 seeds, intervals are 95%-CIs.)

doing full language modelling. As these structural tasks are more challenging, the models do not achieve perfect generalization performance to the held-out test set in every case, which allows us to assess the impact of imperfect generalization on the results.

We show the basic effect of feature type on representation variance in Fig. 9. At the penultimate decoder layer, substantially more variance is explained by the simplest, non-structural, feature compared to the features involving structure. At the encoder layer, however, the pattern is nearly the opposite—most variance is explained by a structural feature (the maximum nesting depth of the sequence). Though the task formulation is different, this finding is broadly in keeping with prior results about how early layers of transformers tend to strongly represent depth in these structures (Yao et al., 2021; Friedman et al., 2023a). Overall, we again observe substantial biases in the representations of the features, but the nature of those biases shifts across the components of the model. Nevertheless, even aggregating across different components of the model, there are clear biases towards some features over others.

In Appendix B.9 we present some supplemental analyses of these results, including showing that the representation variance explained by each feature across seeds does not appear to be strongly correlated with the test accuracy of that feature.

## 4.2 More language-like datasets reveal an additional output-range bias

We next evaluated transformers on a set of datasets that are more like natural language: sentences sampled from a grammar that yields real English sentences with complex syntactic structures (center embedding or

sequential dependencies), and semantic properties (such as valence and noun properties). See Appendix A.2.2 for details. We trained transformers to output various judgements about these sentences, such as identifying the sentence structure, or reporting which noun is the main subject. We analyze the representations with respect to these features.

In Fig. 10 we show the results. There are several interesting patterns. At the final encoder layer, the representations are again dominated by the syntactic feature (sentence structure). However, at the penultimate decoder layer, the variance is dominated by two features that involve reporting the identity of the nouns that play different roles in the sentence. Reporting more general features of these nouns, such as whether they are plural or singular, does not lead to as high of representational variance. Likewise, reporting valence of the sentence (which can be positive, neutral, or negative) carries more variance than reporting whether the sentence is in present or past tense (two values). Thus, the number of possible output values a feature can take may strongly affect the variance its representations carry near the output—which seems reasonable in retrospect, given that each possible value that can be output is effectively a separate output logit. These patterns are also reflected in the encoder, but to a lesser extent. These results show that biases can still persist in slightly more naturalistic settings, and can also be driven by the number of possible outputs associated with a feature.

## 5 Vision

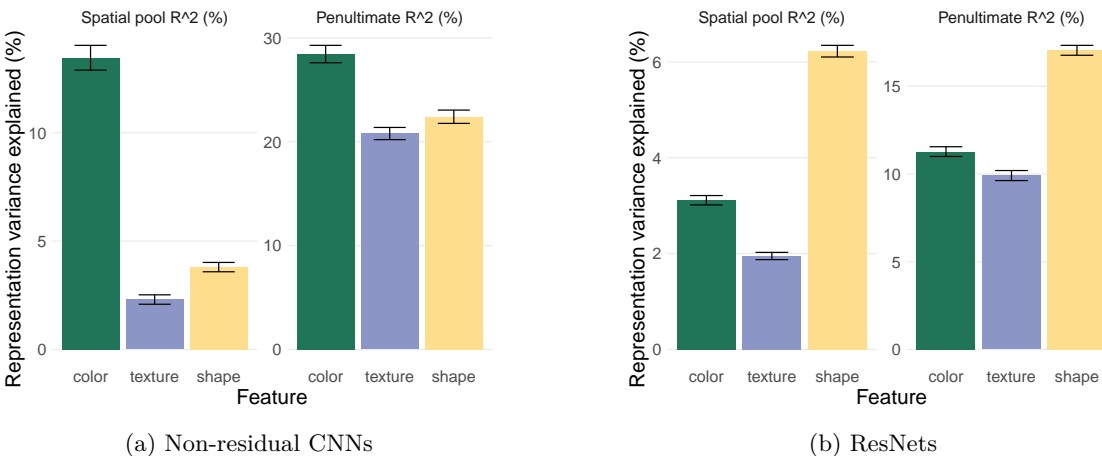

(a) Non-residual CNNs

(b) ResNets

Figure 11: Representation variance explained at the final pooling layer, and penultimate MLP layer for non-residual CNNs and ResNets trained to classify colors, textures, and shapes of objects in images. (a) non-residual CNNs show a strong representational bias towards color—the easiest feature—at the spatial pooling layer, which persists with reduced magnitude at the penultimate layer. (b) ResNets instead show a strong bias towards *shape* representations, which is again somewhat weaker at the penultimate layer. (Bars are averages across 32 seeds, intervals are 95%-CIs.)

We round out our experiments by considering a third large class of problems—computer vision—and some architectures typically used to solve them. Specifically, we construct synthetic datasets of images that show objects with distinct colors, shapes, and textures (examples provided in Appendix A.2.3). We train ResNets (He et al., 2016) and other CNNs without residual connections to classify all of these features simultaneously (architectural details in Appendix A.1). These experiments are again inspired by prior work finding that features like color are easier for models to learn, and will dominate when multiple features compete to predict a label (Hermann & Lampinen, 2020). Here, we investigate the models' internal representations of these features when all features are learned.

We find that non-residual CNNs exhibit a strong bias towards the simplest feature (color, which can be decoded from any single pixel) at the final spatial pooling layer, and a somewhat weaker bias at the penultimate classifier layer (Fig. 11a). This is in keeping with the general pattern observed above, that models

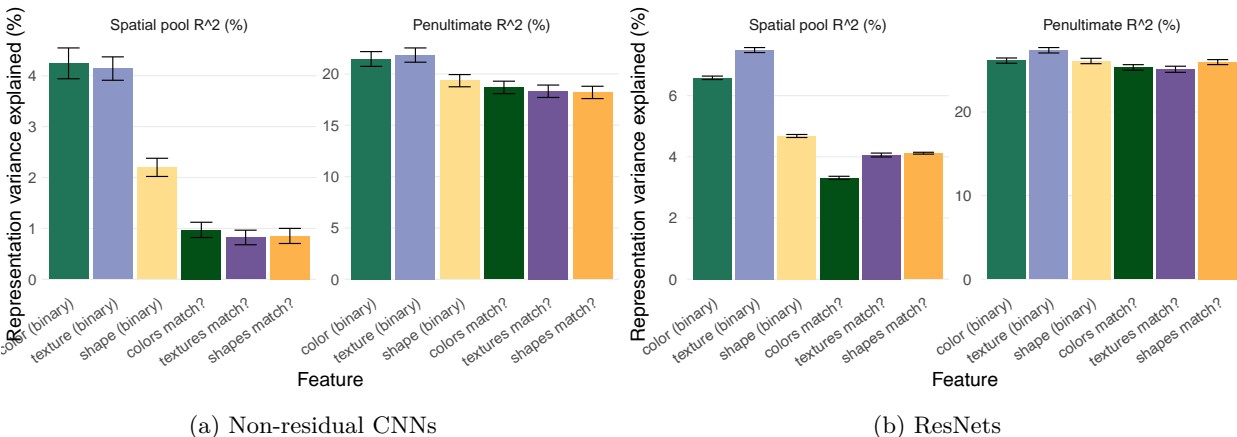

(a) Non-residual CNNs          (b) ResNets

Figure 12: Representation variance explained at the final pooling layer, and penultimate MLP layer for non-residual CNNs and ResNets trained to classify properties of one object, and relations among the properties of the other two, in images with three objects. In this setting, both non-residual CNNs and ResNets show more similar biases, with higher variance explained by the basic features than the relational ones, and higher variance explained by color/texture than shape. Intriguingly, these biases are mostly attenuated by the penultimate layer in this setting, for both model classes. (Bars are averages across 32 seeds, intervals are 95%-CIs.)

are biased towards representing simpler features more strongly, as well as with prior work. Intriguingly, however, ResNets exhibit the *opposite* bias—with representations that strongly favor shape (Fig. 11b). This is despite the fact that color remains the easiest feature, and shape the most difficult (slowest) feature to learn for both models (Fig. 38; as in Hermann & Lampinen 2020).

One possibility is that this bias towards shape may stem from the interaction of the residual connections and the fact that shape requires integrating information over larger areas. Similarly, in the transformer experiments (§4), we noted relatively stronger biases in the *encoders*—which have residual connections to the input—towards features that require integrating over larger input areas. However, we do not see shifts in the easy-hard feature biases of residually connected MLPs (Fig. 27). Taken together, these findings suggest that residual connections across layers may shift the biases of models with respect to extent in space or a sequence. Note, however, that these biases also interact with other features of the input and output structure, as we will see in the next section.

## 5.1 Relational vision tasks show similar complexity biases

All the visual features we considered above involve computations that are arguably simpler (disjunctive template matching) than the relational features (like XOR or grammatical structure) that we considered in our MLP and Transformer experiments. Thus, we performed some experiments where we trained models on images with three objects, and asked them to classify the properties of one object, but to report the *relations* (same or different) among the properties of the other two objects. We reduced the basic category classifications to binary labels (arbitrary splits of each type), to equalize the number of classes of basic and relational categories (as we observed above that the number of classes affects variance).

In Fig. 12 we show the results. As would be expected, the easier basic feature representations tend to carry more variance than the harder relational ones at the final spatial pooling layer in both architectures. Surprisingly, *both* model classes show consistent biases for color and texture over shape, for the basic features. All biases are very weak by the penultimate layer; moreso than we observed above. In general, the biases are weaker in this setting than with more output classes; together with a supplemental experiment (Appendix Fig. 39) showing that binary classifications show a weaker bias in the setting of the original classifications above, these results suggest that feature biases may be stronger with larger output spaces, which accords with our transformer results showing that number of labels itself can be a strong biasing factor.

Why do the basic feature biases of ResNets change direction in these tasks, compared to those above? In follow-up experiments (Appendix B.10.1) we show that switching to binary classification, and the addition of other objects to the image, may both contribute to reversing these biases. These results highlight the complex interactions that affect representational biases. However, the bias towards basic over relational features seems relatively more consistent.

## 6 Why should we care? Evaluating downstream impacts

Computing representational variance explained by features is not a standard analysis. However, the variance explained by the features has various consequences. Some of these consequences are mathematically necessary, and we illustrate them purely for clarity, but some we merely observe empirically. In this section, we illustrate these downstream implications in causal interventions, interpretability, representational similarity analyses, and use of representations for downstream tasks.

### 6.1 The representations identified are causally sufficient and specific

The above analyses are correlational. However, most philosophical accounts (e.g. Cao, 2022; Baker et al., 2022) define representations partly in terms of their causal role in a computation. An increasing number of approaches in interpretability similarly use causal interventions to achieve more faithful interpretations of model internal representations (e.g. Geiger et al., 2021; Meng et al., 2022; Geiger et al., 2024). Here, we verify in one case that the representations identified in our analyses indeed play a causal role in the computations.

To do so, we take an MLP model (from the Fig. 2 experiments) at the end of training, manipulate its internal representations, and evaluate the causal effect on its output behavior.

To manipulate model representations of a feature, we take the representation pattern predicted when that feature is present or absent and subtract those two patterns to create a present-absent difference vector. We then add (or subtract) a scalar multiple of this difference vector to the representation of each stimulus in the test dataset, and evaluate whether by doing so we can flip the label the model predicts for that feature on that stimulus. This approach is equivalent to approaches that create steering vectors from activation differences (e.g. Turner et al., 2023).

We show the results in Fig. 13. If we intervene along the subspace defined by the easy feature difference vector, we can flip the models labels of the easy feature for all items, without altering any of its labels for the hard feature. Similarly, if we intervene along the direction of the hard feature difference vector, we can flip the hard labels without altering the easy ones (unless we make an extremely large intervention). Thus, the representations identified in our regressions are causally sufficient and specific.

Moreover, these results provide another way to see that the hard features occupy less variance than the easy feature — much smaller interventions on the representations suffice to flip the models' hard feature labels. These results might also have downstream consequences for each feature's robustness to perturbation.

### 6.2 Impact on model simplification for interpretability

Many approaches to mechanistic interpretability involve simplifying the model's representations, for example visualizing their top principal components (PCs). However, these simplifications may exacerbate feature biases; if the top PCs are biased towards certain features, lower-variance features may be missed in the interpretation. Thus, these representation simplifications may not faithfully capture the computations of the original model.

We describe one way in which this could occur here, motivated by the experiments of Friedman et al. (2023a). That work shows that certain simplifications may be less faithful out-of-distribution and thus interpretations based on these simplifications may not give generalizable understanding of a model's computations. For example, keeping only the top principal components of a transformer model's keys and queries can yield accurate model behavior in distribution, but fail to generalize appropriately to more challenging test sets,

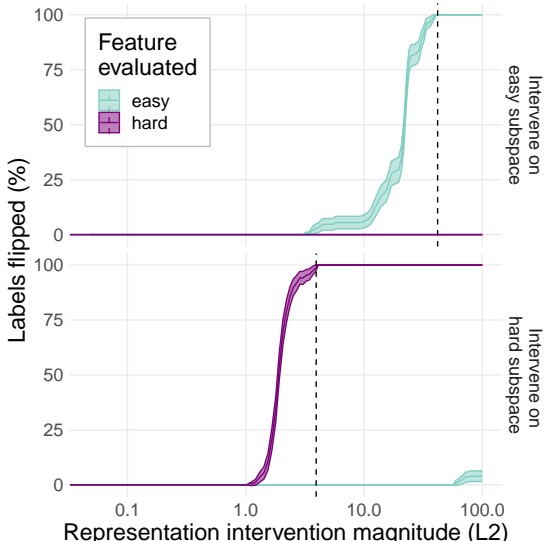

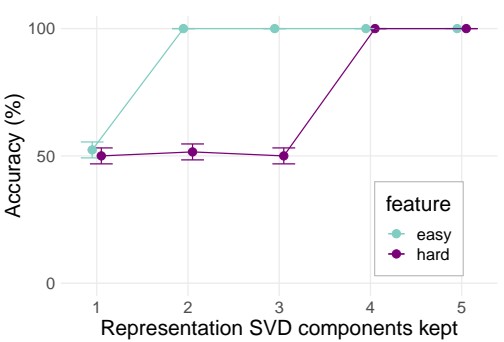

Figure 13: The representations identified by our analyses are causally responsible for the model's behavior. If we intervene along the subspace defined by the each feature, we can flip the model's labels for that feature for every item without altering any of its labels for the other feature. Smaller interventions are sufficient to flip hard feature labels because the hard feature occupies less representational variance. (Vertical dashed line shows the difference vector magnitude from the regression, i.e. the intervention that we expect would flip all labels.)

Figure 14: If we keep only the top-$k$ principal components from a model's representations, the easy feature will be preserved with a smaller $k$ than the hard feature.

suggesting that the original model may use subtler features in more challenging cases, that the simplifications omit.

Though our experimental settings are somewhat different, our results hint at a possible basis for those phenomena. In particular, if the features required to solve the harder OOD splits are more complex, they may occupy relatively less variance, and therefore be more readily lost when keeping only the top principal components. We show a simple demonstration of this from one of our MLP models in Fig. 14: if we keep only two principal components from the models representations, the model can still identify the easy feature perfectly, while hard feature accuracy remains at chance until we keep at least four principal components. (In Appx. B.11 we show that *dropping* the top PCs yields the opposite pattern, with hard features proving more robust).

### 6.2.1 Impact on feature visualization

Given the above observations, clearly low-dimensional visualization may be biased towards certain features. We showed a simple example of this in Fig. 3, in which the top PCs are entirely clustered by the easy feature. In Appx. B.12 we elaborate these results, showing that the hard feature appears in the next PC. However, if models are trained with more and more easy features, the hard feature is pushed into higher and higher PCs. Visualizing with $t$-SNE yields somewhat better results with only a single easy feature—due to $t$-SNE's nonlinear embedding and emphasis on neighbors—but once enough easy features are added it similarly becomes difficult to identify the hard feature in low-dimensional $t$-SNE plots as well, while the easy features show extremely clear clusters. Thus, feature visualizations may be inherently biased towards simpler features, those that are learned earlier, etc.

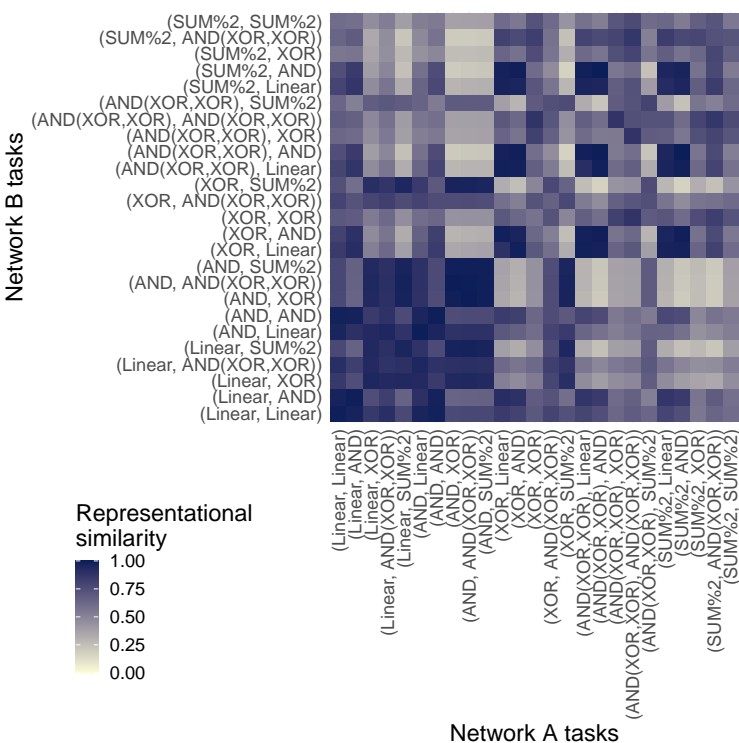

Figure 15: Representational similarities among MLPs trained to compute different pairs of features. The feature representation biases produce strong biases in which networks appear similar; in some cases, networks may appear more similar to ones that are computing entirely different features than they do to another network computing the same features. (Results are computed across 5 seeds per feature pair, using Euclidean distances and Pearson correlation.)

## 6.3 Impact on Representational Similarity Analysis

Representational Similarity Analysis (RSA; Kriegeskorte et al., 2008) is a technique for comparing two systems (e.g. a model and the brain) based on the (dis)similarity structure of their internal representations. The basic idea is that while two systems' representations are not directly comparable, their similarity structures are. RSA and related techniques (e.g. CKA; Kornblith et al., 2019) are commonly used in neuroscience to compare models to brains, and in AI for model analysis (Sucholutsky et al., 2023). However, several prior works have noted that RSA may not perfectly reflect computational similarity (Maheswaranathan et al., 2019; Hermann & Lampinen, 2020; Dujmović et al., 2023; Friedman et al., 2023b); our findings likewise highlight the potential for representational similarity to dissociate from computational similarity.

To illustrate this, in Fig. 15 we perform an RSA analysis between many pairs of MLP networks. Each MLP is trained to output two features of varying difficulty. Ideally, the RSA would be more similar on the diagonal (for networks trained to perform the same two tasks) than off the diagonal, and those on the block diagonals (trained with just one feature that is the same) would be more similar than those off. However, the feature representation biases lead to a substantially more biased structure. For example, two models that are both computing SUM%2 over two sets of features will appear to be *less* similar to one another than each appears to be a network that is *solely* computing simple linear features. These results use a Euclidean metric for RSA, which is naturally sensitive to variance. However, in Appx. B.13 we show that the results are qualitatively similar with cosine distance as the metric.

These results highlight how RSA may be misleading when representations are driven more by some features than by others.

## 6.4   Impact on downstream use of representations

A key motivation for representation learning is to use the representations to help solve downstream tasks. Here, we assess how feature biases in the pretrained representations will affect the behavior of the downstream model.

To do so, we take several MLP models trained above, and train (non)linear classifiers on top of their representations, to predict a label that is probabilistically related to *both* easy and hard features independently. More specifically, we create a range of datasets where the label is independently predicted by the easy and hard features with varying probability, but where both features are equally predictive. We then test the extent to which the classifiers rely on each feature by testing on datasets where the features are decorrelated, and only one feature predicts the label. This setting is similar to some of the experiments of Hermann & Lampinen (2020) and Hermann et al. (2023), but beginning from multi-task pretrained representations rather than raw inputs.

In this setting, we show in Fig. 16 that the biases in the feature representations affect the downstream model behavior. Specifically, we plot the degree to which the classifier outputs "prefer" each feature when they are pitted against one another, across values of feature predictivity. If the features have low predictivity, the classifier relies equally on both features. However, as the features' predictivity increases, the classifiers exhibits increasing bias towards the easier feature.

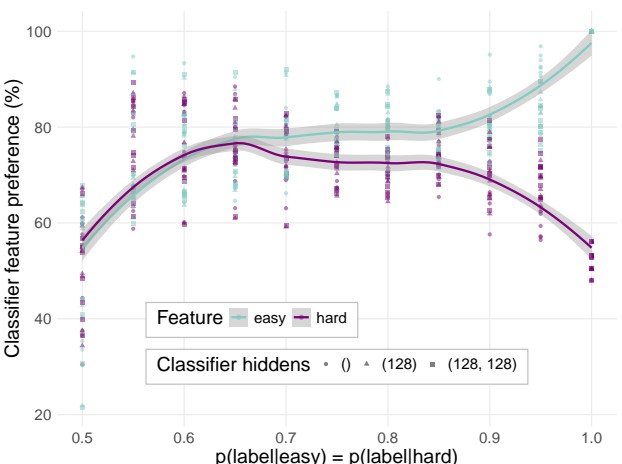

Figure 16: Representational biases affect the generalization behavior of downstream ML models that build on the representations. This figure shows the results of training classifiers (linear or nonlinear) on top of the representations of an MLP model to predict a feature that is stochastically related to both the easy and hard features. If both features are highly predictive, the classifiers are strongly biased toward relying on the easier feature. (Results are similar for both linear and nonlinear classifiers, so we collapse across them.)

Thus, these biases in feature representation, even in a model that learns both features successfully, will affect how downstream models behave.

# 7   Related work

**Implicit inductive biases towards simplicity:** Highly over-parameterized deep networks sometimes generalize well even when they could memorize the entire dataset (Zhang et al., 2021). To make sense of these observations, a variety of works have proposed an implicit inductive bias for simplicity, either in the architectures themselves (Valle-Perez et al., 2018; Huh et al., 2021; Rahaman et al., 2019) or in combination with gradient-based learning dynamics (Tachet et al., 2018; Arora et al., 2019; Advani et al., 2020; Kalimeris et al., 2019; Lampinen & Ganguli, 2018; Shah et al., 2020; Lu et al., 2023; Xu et al., 2023). In particular, Valle-Perez et al. (2018) emphasize that networks with random parameters are biased towards computing simpler functions, which may contribute to the relative ease of learning simpler features—and the greater sparsity in representations of more complex features.

However, most of these prior works have focused on the case where the features redundantly serve a single task, rather than the case where the network is computing all of them, and have measured this bias at the level of the input-output behavior, rather than in the network's internal representations. Our results suggest that the impact of these biases persists in the network's internal representations even in a setting where it has learned to compute the more complex features equally well. Moreover, these representational biases may even help to amplify other inductive biases because, once some features are more strongly represented, they will be favored in later layers' learning (cf. §6.4); thus, they may be part of the mechanism of the

broader phenomenon, rather than just a consequence. However, there are also cases—like the shape bias in ResNets trained on single-object images, despite color being learned first—in which the behavioral and representational biases follow distinct trajectories over learning.

**Representational biases towards simple, common features:** Several other empirical works have noted that representations can be biased towards capturing simpler features and suppressing irrelevant (or more complex) ones, including Hermann & Lampinen (2020) and more recent work on contrastive learning (Xue et al., 2023), and boolean feature learning (Qiu et al., 2024). Other works have noted feature-level biases in which image features are easiest to learn (e.g. Dasgupta et al., 2022). In addition, several works have noted biases due to prevalence, e.g. that learned image features focus more on prevalent features (Benjamin et al., 2022), cosine similarity of word embeddings is biased by prevalence (Zhou et al., 2022), or that pruning selectively harms performance on rare exceptions within a category (Hooker et al., 2019). A more recent study by Morwani et al. (2023) suggests that some representational biases may stem from networks learning maximum-margin solutions. One earlier work on modularity (Csordás et al., 2021) noted in passing some biases in the proportion of network *weights* devoted to two different tasks. However, aside from that, most works have focused on the case where multiple features are used to predict the same label, and thus there is effective competition between features. Here we characterize these biases more systematically, in the case that all features are playing an equally-important, independent computational role. Moreover, we go beyond complexity to characterize other biases, such as the output position biases in transformers.

A concurrent work from Fel et al. (2024) studies the features learned by ResNets trained on ImageNet (Deng et al., 2009), and finds broadly compatible patterns: simpler features tend to be learned earlier than more complex ones, and simpler features also tend to be more important to the network's decisions—but complex features do still play a role. These results complement ours by exploring a more realistic setting, which correspondingly implies less systematic control over the task or the statistics of the input and input-output distributions.

**Shortcut learning:** A variety of works have studied the problem of shortcut learning (Geirhos et al., 2020); i.e., that networks may learn to utilize spuriously-correlated features in the training dataset, and thus fail to generalize to out-of-distribution tests. For example, ImageNet models are often biased towards using texture over shape (Geirhos et al., 2018), and this bias is shaped by the choices like augmentations typically used in training (Hermann et al., 2020). Similarly, a variety of works have shown that (small) language models often rely on semantic heuristics rather than more complex syntactic computations (McCoy et al., 2019; 2020). In many cases these shortcuts appear simpler than the true solution, which suggests that they might arise similarly from the inductive bias towards simplicity (cf. Shah et al., 2020). Indeed, Hermann et al. (2023) show that models are biased towards preferring features that are more *available* (e.g. have greater spatial extent). However, again most of these works have considered settings where the features are both relevant to a single task, rather than serving independent computations.

**Learning disentangled representations:** One line of work focuses on learning *disentangled* representations of inputs (Higgins et al., 2017a;b; Träuble et al., 2021; Bricken et al., 2023a), either to improve learning, generalization, or interpretability. Disentangled representations can improve results in some cases (Van Steenkiste et al., 2019), but may not always (Montero et al., 2020; 2022); again, because the relationship between representation and computation is nontrivial. However, these disentangling objectives likely would shift the feature-level biases we consider.

**Challenges of interpretability:** Various priors works have noted challenges to interpretability methods (Adebayo et al., 2018; Ghorbani et al., 2019; Jain & Wallace, 2019; Wiegreffe & Pinter, 2019; Adebayo et al., 2021; Bolukbasi et al., 2021; Zhou et al., 2022; Friedman et al., 2023a;b; Wen et al., 2023), often suggesting that saliency or activations can be misleading about what a model is "doing" computationally. Our results highlight similar challenges, that perhaps underpin some prior observations. For example, Friedman et al. (2023a) noted that simplifying model internal activations using techniques like PCA could yield interpretations that are less faithful on more challenging test distributions; as we noted above (§6.2), our results may point to a possible explanation. More generally, many interpretability methods are implicitly predicated on the idea that variance in the neural representations effectively conveys importance; our results imply that these methods may be similarly biased.

**Analyzing and interpreting representations on algorithmic tasks:** In order to circumvent the challenges of interpretability in complex settings, a variety of prior works have analyzed the representations and circuits of models on controlled, algorithmic settings where the computational goal of the model is clearly specified. For example, these tasks include modular arithmetic (Nanda et al., 2023; Liu et al., 2022; Zhong et al., 2024) and Dyck languages (Hewitt et al., 2020; Yao et al., 2021; Murty et al., 2023; Friedman et al., 2023a). These algorithmic tasks reveal rich learning dynamics in which more complex, generalizing features are often learned later in a "grokking" stage (Power et al., 2022)—likely because the complex features that generalize take longer to learn (Xu et al., 2023). However, like the above works on representational biases, most of these works have studied the case where features are redundantly predictive of a single task to differing degrees, often have confounds of complexity with utility, and have focused on a small range of tasks. Our work complements these by performing controlled studies of representation biases across a wider range of tasks and architectures, in cases when the features being represented are independently useful rather than redundant or confounded—but correspondingly at the cost of performing less in-depth analysis of the circuitry for solving any particular problem than in prior works.

**Neuroscience:** Issues like those we describe here are not unfamiliar in neuroscience; it is commonly acknowledged that making inferences from representation to computation is challenging, even with interventional methods (e.g. Jonas & Kording, 2017). For example, it is well-known that neural signals can be strongly driven by "simple" or otherwise-salient variables like movement (Musall et al., 2019), which can interact with the representation of task-relevant variables in complex ways. More generally, representations can be biased towards certain features—such as biases towards horizontal and vertical orientations in primary visual cortex (e.g. Levick & Thibos, 1982). These biases can yield suboptimal, biased performance in downstream behavioral tasks, that appear to be driven by learning via feature reweighting rather than feature tuning (Laamerad et al., 2024)—which is also one explanation for why linear features are easier to learn than nonlinear ones in our setting. Other works have highlighted the potential for aliasing between systems computing different functions when using RSA over limited datasets (Dujmović et al., 2023), and conversely representational divergences among different architectures computing similar things (Maheswaranathan et al., 2019). Complexities like these led Kriegeskorte & Diedrichsen (2019) to highlight the need to "peel away layers that reflect developmental coincidences, random biological variation, and other epiphenomena without functional relevance, to arrive at the brain's representational core: the neural code used by the brain itself to mind and manipulate the world." We hope our analyses help to provide inspiration for understanding some of these complexities and coincidences, and more pragmatically the issues that may arise in brain-model comparisons.

# 8 Discussion

We have attempted to systematically characterize the internal representations of deep learning models that are trained to compute multiple features of their input. We found that these representations can be biased towards one feature or another by a variety of factors such as the relative complexities of the features, their positions and extents, their prevalences, and their learning order. Some of these biases interact with architecture and optimizer in complex ways, while others are relatively consistent.

We believe characterizing these biases is valuable for multiple reasons. First, there is inherent scientific value in exploring the complex phenomena of gradient-based representation learning that are not, to the best of our knowledge, fully understood. Some of our observations echo the previously-observed simplicity bias of deep learning architectures, but at a representational rather than behavioral level. These representational simplicity biases appear to be partly driven be learning order, and partly by the fact that there are inherently more ways to represent a more nonlinear feature. Second, our findings have important practical implications for interpretability and neuroscience methods that rely on analyzing model representations.

**Implications for interpretability:** Machine-learning researchers often want to interpret model representations to understand the models (Olah et al., 2018; Geiger et al., 2021; Olsson et al., 2022; Bills et al., 2023)—either for scientific reasons, or to attempt to assess their safety and reliability, or to improve model behavior (Zou et al., 2023; Muttenthaler et al., 2024). However, many of these analyses implicitly or explicitly prioritize features that carry high variance in the representations. This prioritization is explicit in the case of using methods like PCA to analyze model representations. Indeed, as we suggested above, our

results might help explain the fact that simplifying models to interpret them, using methods like PCA, may not generalize as well out of distribution (Friedman et al., 2023a). However, a variety of other analyses also implicitly prioritize high-variance components, and may therefore likewise be subject to some bias.

For example, several recent works have proposed to tackle superposition in model activations via sparse dictionary learning (Bricken et al., 2023b; Cunningham et al., 2023). These methods use a sparse autoencoder (SAE) with a high-dimensional latent space and a sparsity penalty to process the internal representations, which can decompose them to yield more interpretable features. However, because the autoencoder is trained to reconstruct the internal representations via an $\ell^2$ loss, the objective favors reconstructing high-variance features; thus, these interpretability methods will likewise potentially be biased towards particular features and may neglect others that nevertheless play an important role in a model's computations.

**Implications for computational neuroscience:** Our findings may also be relevant to computational neuroscience. Research in this area has increasingly found correspondences between representations in deep learning models trained for vision or language processing, and representations in corresponding brain regions in humans (Yamins et al., 2014; Yamins & DiCarlo, 2016; Schrimpf et al., 2018; 2021; Tuckute et al., 2023; Hosseini et al., 2024). Yet, as models have come to be trained on increasingly large datasets, the results have become perplexing. Conwell et al. (e.g. 2023) note that seemingly-diverse classes of models often end up achieving similarly high degrees of ability to predict (regress) brain representations, even when their representational structure is different—either under RSA or more conservative comparison metrics (Khosla & Williams, 2023). Our findings hint at several potential explanations of this phenomenon: if the models differ in representational biases (e.g., if some are more biased towards easy features than others), but ultimately compute the same functions, their representational similarity structure might be dissimilar for reasons that are essentially orthogonal to the computations they perform.

More speculatively, if the *brain* representations are *also* biased towards representing simpler features more strongly, then explaining most of the variance in brain representations might not be much of an achievement— the most complex, interesting signals might be in the low-variance components that both regressions and RSA will tend to neglect. However, neural coding in the brain is likely optimized via other learning rules, and for other factors (like energy efficiency; Laughlin, 2001) that may shift its biases.

In either case, representational biases in the systems we are analyzing may lead us to make biased inferences about their relationship at an algorithmic or computational level—either under- or over-estimating their similarity.

**Levels of analysis:** Marr (1982) proposed three levels of analysis for a system (in neuroscience): the computational level (the problem a system is trying to solve), the algorithmic level (the approach to solving it), and the implementational level (how that approach is implemented on the hardware). Prior work has highlighted the complex relationships between these levels of analysis (Churchland & Sejnowski, 1990; Shea, 2018). Similarly, our results both illustrate the separation between these levels—for example, the fact that there are many representation structures that realize the computation of the feature SUM % 2—but also their complex interactions. For example, the representational biases shift which features a model will use for a downstream classification task, thus illustrating how representation structure can influence higher levels of abstraction like the function a model will learn to compute.

It is natural to ask whether our terminology fits these levels of analysis appropriately. In our motivation, we described the different features as playing the same "computational" role, in the sense that the system is trained to output all of them. However, it would be reasonable to say that feature complexity, for example, changes the "computational" role that a feature plays in a model—it depends on where one draws the boundary between the computational and algorithmic. We have adopted a particular convention for clarity, but we do not think that adjusting this boundary and our terminology would fundamentally change our results or their implications.

**Do interventional methods address these challenges?** Sexton & Love (2022) likewise note that variance in representations does not always capture computational importance. The authors therefore propose more directly computational comparisons: replacing the representations in one system (a model) with representations regressed from another system (a brain). This method does indeed offer much stronger constraints

on the system that is intervened upon, which allows for novel insights. However, knowing whether a model has captured *all* the computationally-important features in the neural data would require intervening on the brain instead—which is usually infeasible. Otherwise, the model might be missing some low-variance features that are nevertheless essential for downstream computations.

The work above is one instance of many interventional methods that have been applied to analyze the computations of deep learning models (Geiger et al., 2021; Olsson et al., 2022; Geiger et al., 2024; Nanda et al., 2023; Conmy et al., 2023; Wu et al., 2024; Shah et al., 2024) and brains (Deisseroth, 2015; Liu et al., 2012). Again, by intervening, these methods do yield stronger constraints, and allow stronger causal interpretations of which features are playing a particular role in a computation.

However, in practice, the hypotheses tested via these more rigorous interventional methods often originate from correlational analyses of representations (e.g. seeing that certain neurons are highly responsive to a feature) or prior knowledge about a task (e.g. computational hypotheses about how the problem should be solved). Our experiments suggest that the first class of approaches may be biased in the kinds of causal hypotheses they generate, e.g. towards hypotheses about simpler features. Similarly, we may believe algorithmically that some features should be used to solve a task, and find representations that correlate with the feature and causally influence the models representations. However, in either case, it will be difficult to rule out the possibility that other subtle representational features are influencing the system's computations in important ways. Even verifying causal sufficiency, as in §3.3, does not eliminate the possibility that some other subtle feature is playing a role in some cases, but is preempted (Mueller, 2024) by our strong intervention on the simpler feature.

Furthermore, our analyses in Section 3.3 suggest that to fully identify the contribution of a feature that is *nonlinearly* involved in the network's computations, we need to account for all the ways that feature could be represented—even with prior knowledge about which features are relevant to a task, this will be challenging in the general case. Thus, while causal methods are critical to thoroughly testing hypotheses about representation and computation, we do not believe they fully resolve the challenges raised here.

**Do our findings allow us to correct for representational biases?** In the case of the simple MLP networks, we were able to "explain"—i.e., design interventions that eliminate—the representational biases we observe. Unfortunately, these interventions required fundamentally altering the training process (to train on *only* the hard feature at first) and also characterizing the representations in terms of *all* input patterns relevant to the hard feature. Each of these would be challenging or impossible to scale to analyzing a larger-scale system like a language model or a brain. Furthermore, they rely on knowing *a priori* exactly what features a system is computing, which is true by design in our work, but not more generally. Moreover, the complex interactions we observe between different features that are trained would make it exceedingly difficult to know what kind of biases to expect in any system that computes a complex distribution of tasks over rich inputs. Thus, while our experiments highlight the existence of these biases, it is not immediately clear how these observations can be directly used to improve existing representation analyses.

**Biases do not mean representation analyses are hopeless:** While some of this discussion may seem pessimistic, the fact that biases might impact our analyses do not mean that those analyses are useless. The high-variance features clearly *are* playing an important role in the models computations, they are just not the only features to do so. Thus, biases in the analyses do not invalidate their conclusions; instead, the biases may *limit* those conclusions, and increase the challenge of gaining *complete* understanding of a system.

## 8.1 Limitations & future directions

There are a number of limitations to our work that would be useful to address in the future. First, we focused on simple, synthetic datasets. While this choice allowed complete control over the tasks, the features, and their correlations, it is possible that biases would be different on more realistic datasets with larger sets of entangled features. However, concurrent work from Fel et al. (2024) finds compatible results for larger models trained on ImageNet.

Moreover, we focused on classification because, again, it made the computational solution more straightforward to analyze. It would be interesting to consider how these biases differ under other objectives—it is

possible that continuous functions yield qualitatively different feature learning than discrete classifications,[6] though an increasing amount of work has converged on some kind of (sequential, soft) classification, e.g. language modeling. However, a limitation of our tasks relative to language modeling is that the representations we consider are not richly contextual. Transformers can learn to restructure feature representations according to the task context (Li & McClelland, 2023); in future work it would therefore be important to study how representation biases interact with more richly contextual computation.

Moreover, the models we used were of the toy scale used in a range of mechanistic interpretability work (e.g. Nanda et al., 2023; Yao et al., 2021). While we did not see dramatic effects of parameter count in the regimes we considered, we did observe somewhat larger biases in deeper models, and it is possible that qualitatively different biases would emerge in larger models; thus it would be useful to study these effects at scale in future work. Furthermore, we found that the biases we observed can be shifted in nontrivial ways by hyperparameter choices (§3.6). However, we did not observe these interactions in Transformers trained on more complex datasets. We therefore expect that in standard training settings, where models are trained on complex tasks without full convergence, the representational biases would be more similar to those observed in earlier stages of training. Nevertheless, future work should investigate these interactions.

Finally, our main analyses relied on linear regression. While these analyses can account for many forms of representation—particularly in §3.3, where the augmented all-input-patterns space can account for any function of the hard inputs alone—they are limited to information that is not linearly available from that augmented space. For example, if the hard features were represented in superposition (cf. Elhage et al., 2022) with other noise inputs, a linear analysis from solely the hard feature space could not account for that. In general, it is difficult to completely test the possibility that some representation non-linearly encodes some information (if it were easy, then encryption would not work). However, the fact that we observe that the majority of the total variance in the system is explained by these linear analyses, and that they are sufficient for causally changing the behavior of the system in a targeted way (§6.1), supports the idea that linear analyses provide useful insight.[7] Finally, linear analyses are empirically justifiable in the context of their frequent use in interpretability and model steering research (e.g. Liu et al., 2022; Turner et al., 2023; Oikarinen & Weng, 2024), and more general arguments that information tends to be represented linearly (Park et al., 2024). Nevertheless, more deeply exploring the space of nonlinear representations within models remains an important direction for future work.

## 9 Conclusions

We have trained deep learning models to compute multiple features of their inputs. We found that, even once all features have been learned, the models' representations are substantially biased towards or away from features by properties like their complexity, the order in which they were learned, their prevalence in the dataset, or their position in the output sequence. These representational biases may relate to some of the implicit inductive biases of deep learning. More pragmatically, representational biases could pose challenges for interpreting learned representations, or comparing them between different systems—in machine learning, cognitive science, and neuroscience.

## Acknowledgements

We thank Roma Patel, Aaditya Singh, Spencer Frei, Dan Friedman, Lukas Muttenthaler, Asma Ghandeharioun, Ed Grefenstette, and the anonymous reviewers for comments and suggestions on earlier versions of the paper.

---

[6]Note, however, that discrete features as input and output can nevertheless be best served by more continuous internal representations. For example, the nonlinear hard feature we use in the first experiments is naturally computed by first summing 4 binary values, giving a sum between 0 and 4 as an intermediate step.

[7]Though note that just observing the causal sufficiency of linear features does not rule out the possibility that some nonlinear feature is playing a role that gets preempted by the intervention on the linear feature (cf. Mueller, 2024).

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

# A  Supplemental methods

All models were trained in JAX (Bradbury et al., 2018) using the flax (Heek et al., 2023) library; networks were generally constructed from the corresponding flax examples. All representation regressions during training were fit using scikit-learn (Pedregosa et al., 2011). Plots were created with the Tidyverse (Wickham et al., 2019). Plot color palettes are loosely based on Harrower & Brewer (2003).

**Demonstration colab:** We provide a simple colab demonstrating the basic feature complexity result at `https://gist.github.com/lampinen-dm/b6541019ef4cf2988669ab44aa82460b`

## A.1  Hyperparameters & architectural details

In Table 1 we report hyperparameter values for all main experiments. Note that different choices for some hyperparameters for the MLP experiments are systematically tested in Section B.7, for example a larger set of optimizers. Hyperparameter values were generally chosen from prior work, or from the flax (Heek et al., 2023) examples; they were changed if the models exhibited training instability or failed to learn.

| Model | Experiment | Optimizer | LR | Train set size | Batch size | Max epochs |
|---|---|---|---|---|---|---|
| MLP | Basic | SGD | $1 \cdot 10^{-3}$ | 4096 | Full | $5 \cdot 10^5$ |
| | Prevalence | | | 32768 | 512 | |
| Transformer | Letter sequence | AdamW | 1e-5 | 16384 | 512 | $10^4$ |
| | Language | | | 32768 | | |
| | Dyck | | 3e-5 | 131072 | | $10^5$ |
| CNN | Basic | Adam | 3e-4 | 8192 | 256 | $10^3$ |
| | Relational | | 3e-5 | 32768 | | |
| ResNet | Basic | | 3e-4 | 8192 | | |
| | Relational | | 1e-4 | 32768 | | |

Table 1: Hyperparameters for main experiments. (Where a value is omitted, it is the same as the value above.)

Additional architectural details are provided below for each model.

**MLP:** The network contained four hidden layers, with sizes [256, 128, 64, 64], followed by a final output layer. Initialization was variance-scaling with scale 1. The nonlinearity at the hidden layers was a leaky rectifier with negative slope 0.01. The model was trained using a sigmoid cross-entropy loss.

**Transformer:** The network was an encoder-decoder transformer (Vaswani et al., 2017) as implemented in the flax examples, with 4 layers in each component, 4 heads per layer, an embedding size of 128 (and thus Q/K/V dimension of 32), and an MLP dimension of 512. The implementation uses the layernorm on the branch input rather than the residual connection, following Radford et al. (2019). The model uses standard sinusoidal position embeddings. The model was trained using a softmax cross-entropy loss, as in standard language modeling.

**CNN:** The architecture was a standard CNN that alternates convolution with mean pooling. Convolution filter sizes were (6, 4, 3, 2), channels were (128, 128, 128, 256, 256), strides were (3, 1, 1, 2, 1), and pooling sizes and strides were (3, *, *, 2, *). All convolutions and the MLP hidden layer used rectifier nonlinearities. The network was trained using a softmax cross-entropy loss for multi-way classification.

**ResNet:** The architecture was a ResNet-18 as implemented in the flax examples, followed by a two-layer MLP classifier (to match the CNN more closely), with 512 units at the hidden layer.

## A.2  Datasets

In this section we provide further details on the datasets. Except where otherwise noted, all datasets were constructed such that different features were statistically independent, and feature values were balanced within each dataset.

### A.2.1  Boolean

For the Boolean experiments, the inputs were vectors with 64 boolean values, and the outputs were vectors with $\geq 2$ boolean values. Some number of the inputs were devoted to each of the features, while the rest were set randomly. The inputs that were relevant for one feature did not overlap with those relevant for another feature. For our main experiments we redundantly encoded the linear features across four input units, to match the number used for the hardest features; however, in Fig. 29 we show this redundant encoding does not substantially affect the results.

### A.2.2  Language

All language problems were posed as sequence-to-sequence tasks, where an input sequence was provided and the model was asked to output a sequence of classifications of features of that input. As the position of features in the output sequence affects their variance, the output sequence order was randomized for each training run.

**Letter strings:** The datasets and features were similar to the boolean tasks, except for a larger input vocabulary, and the sequential structure of the inputs. The input sentences were constructed from a vocabulary of 9 letters (A-I). There were 5 input letters provided per feature, with the relevant letters appearing at the beginning of that chunk, thus there was at least one irrelevant (noise) letter at the end of each feature's relevant inputs. Since 8 features were used, input sequences were length $8 \times 5 = 40$. The input letters were initially sampled randomly, but then labels were assigned to each input to balance each feature (independently) across the dataset, and the relevant input letters were then resampled until the desired label was reached (rejection sampling). The features used were applied to chunks of 1-4 letters, with the feature being either exact match to a letter string, all-but-one letter matching (XOR-like), or the sum of the letter matches mod 2.

**Naturalistic language:** The sentences used for this work were sampled from a simple grammar that generates grammatical English sentences, but within a relatively constrained range. The grammar focuses on two types of sentence structures—with either nested or successive noun-verb dependencies—which are loosely inspired by prior work on grammar processing in neural models (Lakretz et al., 2021). The primitives of the grammar are provided below; the items with a question mark were introduced with some probability (0.5 for the adjectives, 0.33 for all others). The valence and tense (present or past) of the sentence were set to be the same across all of the adjectives/verbs in the sentence.

```
SENTENCE_STRUCTURES = {
    'sequential': 'the␣(A?)␣(N1)␣(L?)␣(RV)␣that␣the␣(A?)␣(N2)␣(L?)␣(V)',
    'nested': 'the␣(A?)␣(N2)␣(L?)␣that␣the␣(A?)␣(N1)␣(L?)␣(RV)␣(V)',
}

ADJECTIVES = {
    'positive': ['good', 'cool', 'awesome', 'wonderful'],
    'negative': ['bad', 'uncool', 'terrible', 'worthless'],
    'neutral': ['acceptable', 'bland', 'generic', 'tolerable'],
}

LANG_AGENTS = [
    {'singular': 'person', 'plural': 'people'},
    {'singular': 'child', 'plural': 'children'},
    {'singular': 'adult', 'plural': 'adults'},
    {'singular': 'dog', 'plural': 'dogs'},
    {'singular': 'horse', 'plural': 'horses'},
    {'singular': 'owl', 'plural': 'owls'},
    {'singular': 'robot', 'plural': 'robots'},
]

LANG_OBJECT_VERBS = {
    'cook':{
        'conjugations': {'present_singular': 'cooks', 'present_plural': 'cook',
                    'past': 'cooked'},
        'objects': [('pizza', 'pizzas'), ('sandwich', 'sandwiches'),
                ('potato', 'potatoes')],
    },
    'eat':{
        'conjugations': {'present_singular': 'eats', 'present_plural': 'eat',
                    'past': 'ate'},
```

```
        'objects': [('pizza', 'pizzas'), ('sandwich', 'sandwiches'),
                   ('potato', 'potatoes'), ('banana', 'bananas')],
    },
    'run':{
        'conjugations': {'present_singular': 'runs', 'present_plural': 'run',
                        'past': 'ran'},
        'objects': [('race', 'races'), ('marathon', 'marathons')],
    },
    'write': {
        'conjugations': {'present_singular': 'writes', 'present_plural': 'write',
                        'past': 'wrote'},
        'objects': [('book', 'books'), ('poem', 'poems'), ('essay', 'essays')],
    }
}
for _, v in LANG_OBJECT_VERBS.items():
  v['conjugations']['past_singular'] = v['conjugations']['past_plural'] = v['conjugations']['past']

LANG_RELATION_VERBS = {
    'see': {'present_singular': 'sees', 'present_plural': 'see',
           'past': 'saw'},
    'hear': {'present_singular': 'hears', 'present_plural': 'hear',
           'past': 'heard'},
    'know': {'present_singular': 'knows', 'present_plural': 'know',
           'past': 'knew'},
    'think': {'present_singular': 'thinks', 'present_plural': 'think',
           'past': 'thought'},
    'say': {'present_singular': 'says', 'present_plural': 'say',
           'past': 'said'},
}
for _, v in LANG_RELATION_VERBS.items():
  v['past_singular'] = v['past_plural'] = v['past']

BE_VERB = {'present_singular': 'is', 'present_plural': 'are',
          'past_singular': 'was', 'past_plural': 'were'}

LOCATION_CLAUSES = [
    'by the ' + x for x in ['park', 'school', 'house', 'office', 'lake', 'forest'] + [z for y in LANG_AGENTS for z in y.values()]]
```

From these sentences, we trained the model to output a variety of features: whether the sentence was written in present or past tense, the valence, the structure type (nested or sequential), whether each of the nouns was singular or plural, and their identities. The latter features require parsing the sentence to determine its structure, and to identify which noun appears where while ignoring the other noun (and any that occur in distractor location clauses).

**Dyck:** For the Dyck language experiments we used sentences sampled from Dyck-(20,10)—that is, 20 bracket types and a max depth of 10. We also enforced a length limit of 64 tokens. We sampled each sequence one token at a time: first we selected whether it would be an open or close bracket. If the type was unconstrained, we set a probability of 70% that it would be the same type as the previous bracket; e.g., that an open bracket would be followed by another open bracket. This biased sampling yields more interesting structures. If the token was at 0 or max depth, or close to the max length, the type was constrained appropriately to fit the language constraints.

If the token was selected to be an open bracket the bracket class was randomly sampled from the set of 20 bracket classes. If it was a close bracket, it was chosen to match the corresponding opening bracket.

After the first token, when the depth reached 0, the sentence was terminated with 50% probability; otherwise, a new opening bracket was sampled. Thus, this dataset contains sentences of varying lengths.

We trained the model to compute 5 features about the sentence: the class of the first token, the class of the first token that appeared at the max depth of the sentence, the maximum depth itself, the number of root nodes (i.e. opening brackets at depth 0), and the max branching factor (i.e. the max number of opening brackets contained within another pair, or the number of root nodes if it was larger). Note that unlike the naturalistic language tasks above, the sentence length is correlated with various features such as the number of root nodes. Thus we control for length in our analyses.

### A.2.3  Vision

In Fig. 17 we show examples of each of the 10 colors, textures, and shapes that we used in the vision experiments. Note that, while our experiments were inspired by Hermann & Lampinen (2020) we do not use precisely the same feature sets that they did; our textures and shapes are somewhat simpler and the fact that we do not randomly rotate them likely makes the classification problem slightly easier. However, we do not expect this to substantially alter the results.

In Fig. 18 we show example stimuli for the basic and relational tasks—note that in the latter case, the basic features are reported for the object in the left "column" of the image, and the relational features are

reported for the objects on the right. Note that for both basic and relational tasks, all objects are randomly offset within their allotted space.

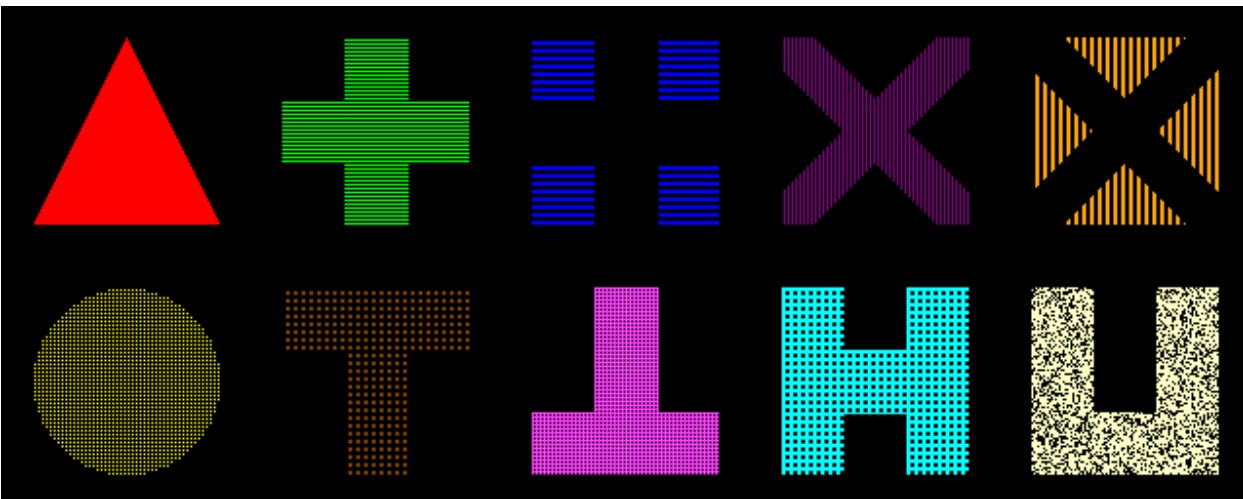

Figure 17: Examples of the 10 colors, textures, and shapes used in the vision experiments.

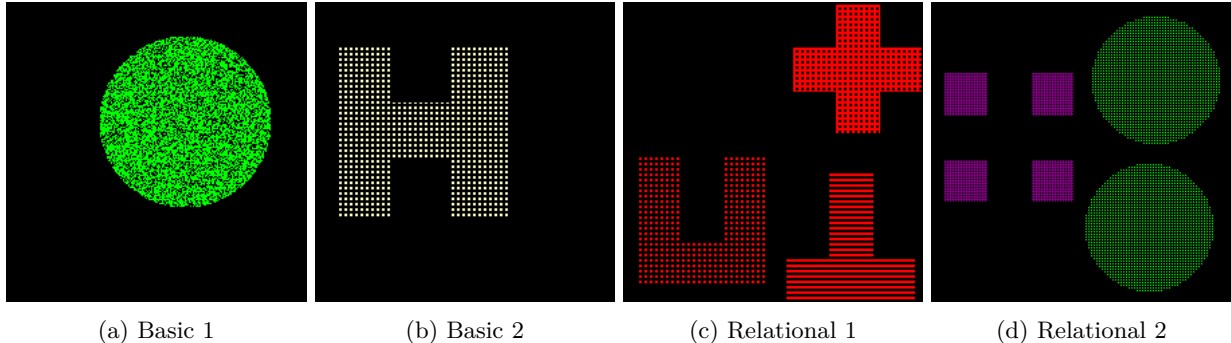

| (a) Basic 1 | (b) Basic 2 | (c) Relational 1 | (d) Relational 2 |

Figure 18: Example stimuli for the vision datasets with single objects (a-b) or for the tasks including relational features among multiple objects (c-d).

## B   Supplemental analyses

### B.1   MLP loss curves

In Fig. 19 we show the feature-wise loss curves corresponding to the base plot in Fig. 2; the losses for both features are trending towards zero.

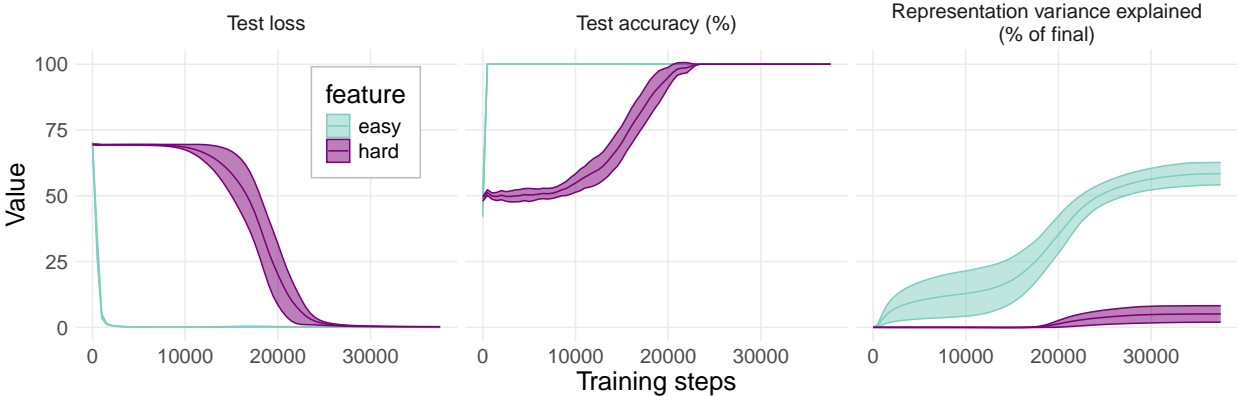

Figure 19: Test losses by feature, as well as accuracies and representation variance explained, for the models plotted in Fig. 2. Both features' losses approach zero as they are learned.

## B.2 Representation variance explained curves with and without normalization

In the learning curves we generally plot variance explained normalized by total variance at the end of training. In Fig. 20 we compare this to the values normalized by current total variance. While the qualitative biases are similar, the alternative normalization makes the earlier learning dynamics more visible. Furthermore, it also shows that at initialization, the easy feature explains more variance than the hard feature—but all hard input patterns explain even more.

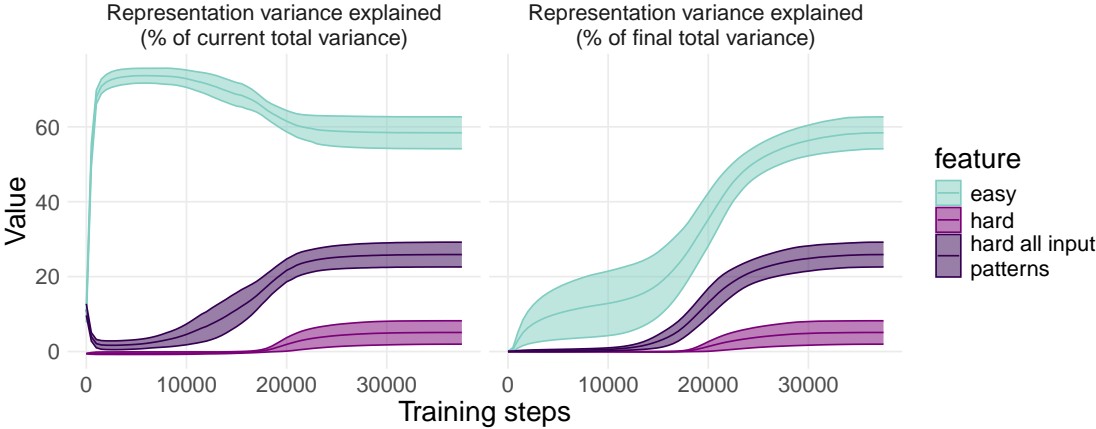

Figure 20: Comparing variance explained as a proportion of total variance at the end of training (right, as plotted elsewhere in this paper) vs. normalized by current total variance; biases are qualitatively similar, but the alternative plot makes the initial learning dynamics clearer.

## B.3 MLP learning curves by training order

In Fig. 21 we show the learning curves for easy and hard features, across different training orders. When the hard feature is pretrained, it occupies more variance than the easy feature initially, but the easy feature quickly overtakes it once it starts to be learned. However, when considering all input patterns associated with the hard feature, the effect of pretraining is much clearer; and the curve more clearly shows the inflation in the later phase of learning that is observed with the easy feature in the other cases.

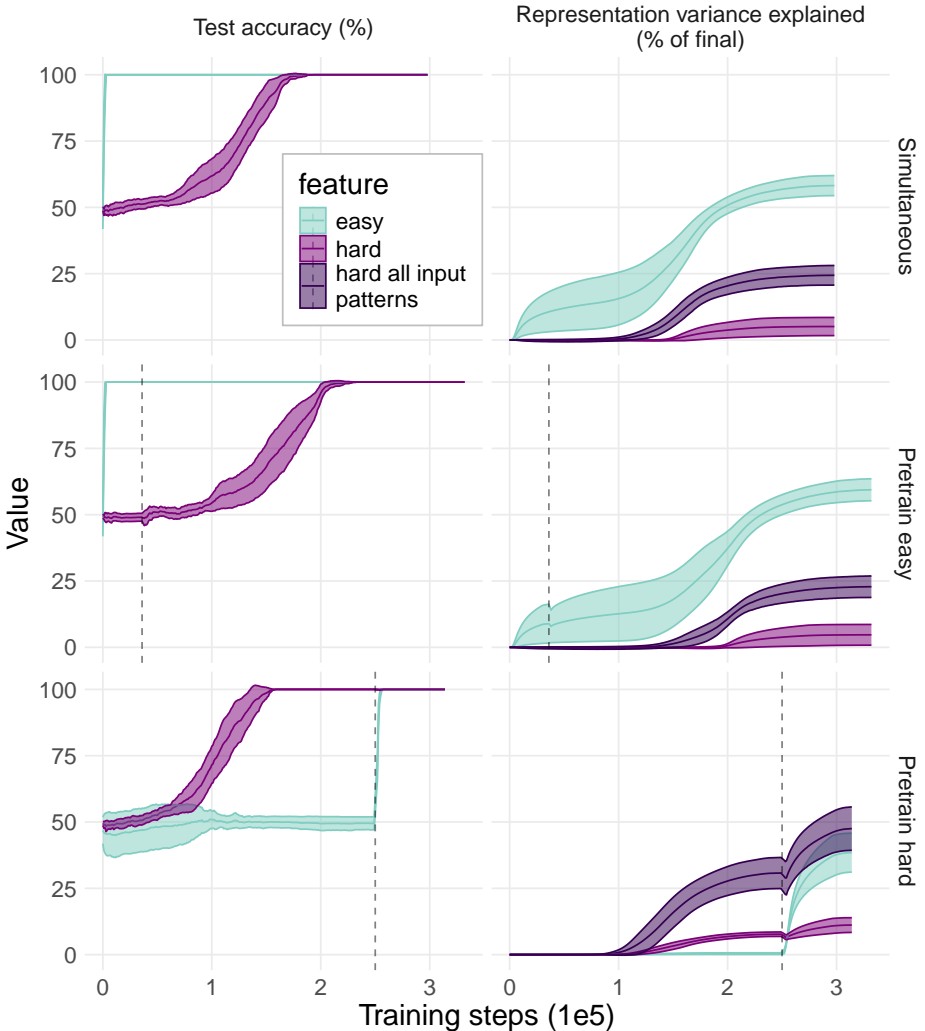

Figure 21: Learning curves for MLPs learning easy and hard tasks, depending on training order—simultaneous training, or pretraining one task. Left panels show test accuracy, right show variance explained in the penultimate representations. (Lines are means across 10 seeds, areas are 95%-CIs).

### B.4 Per-unit representation variance analysis

In this section we present analyses of how each of the 64 units in the penultimate layer of the model represents each feature for a single run of the basic model trained on the linear and sum mod 2 tasks. In Fig. 22 we show snapshots of the variance explained by each feature for each unit, at various salient stages in training. In Fig. 23 we show individual learning curves for all the units. The qualitative patterns are similar to the aggregate case, but there are interesting individual effects to observe. Most notably, the units that end up carrying the most variance about the difficult feature are also those that initially carry the most variance about the easy feature, and the variance they carry about the easy feature is inflated as the hard feature is learned. Thus, most of the individual units also continue to carry more variance due to the easy feature. Only a subset of the initially-easy-selective neurons acquire some hard-relevant signal later (which likely contributes to the sparsity difference observed below). However, a few neurons that initially carried at least as much signal about the hard inputs as easy ones do tend to become somewhat more selective for the hard input features as the harder task is learned. Furthermore, many of the units carry non-trivial signal about other inputs (or nonlinear patterns of the feature-relevant inputs) that is *also* amplified as the other features

are learned — either because those inputs are weakly correlated with a task feature within the dataset by chance, or just due to the learning dynamics.

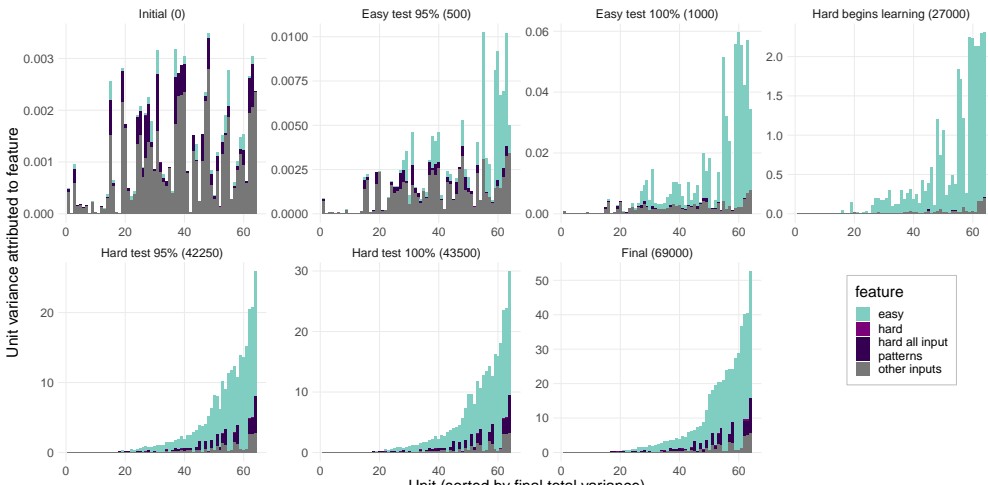

Figure 22: The distribution of variance explained by each feature across the 64 units of an MLP at various stages of training. (The ordering of units is consistent across the plots, and is determined by the final plot.)

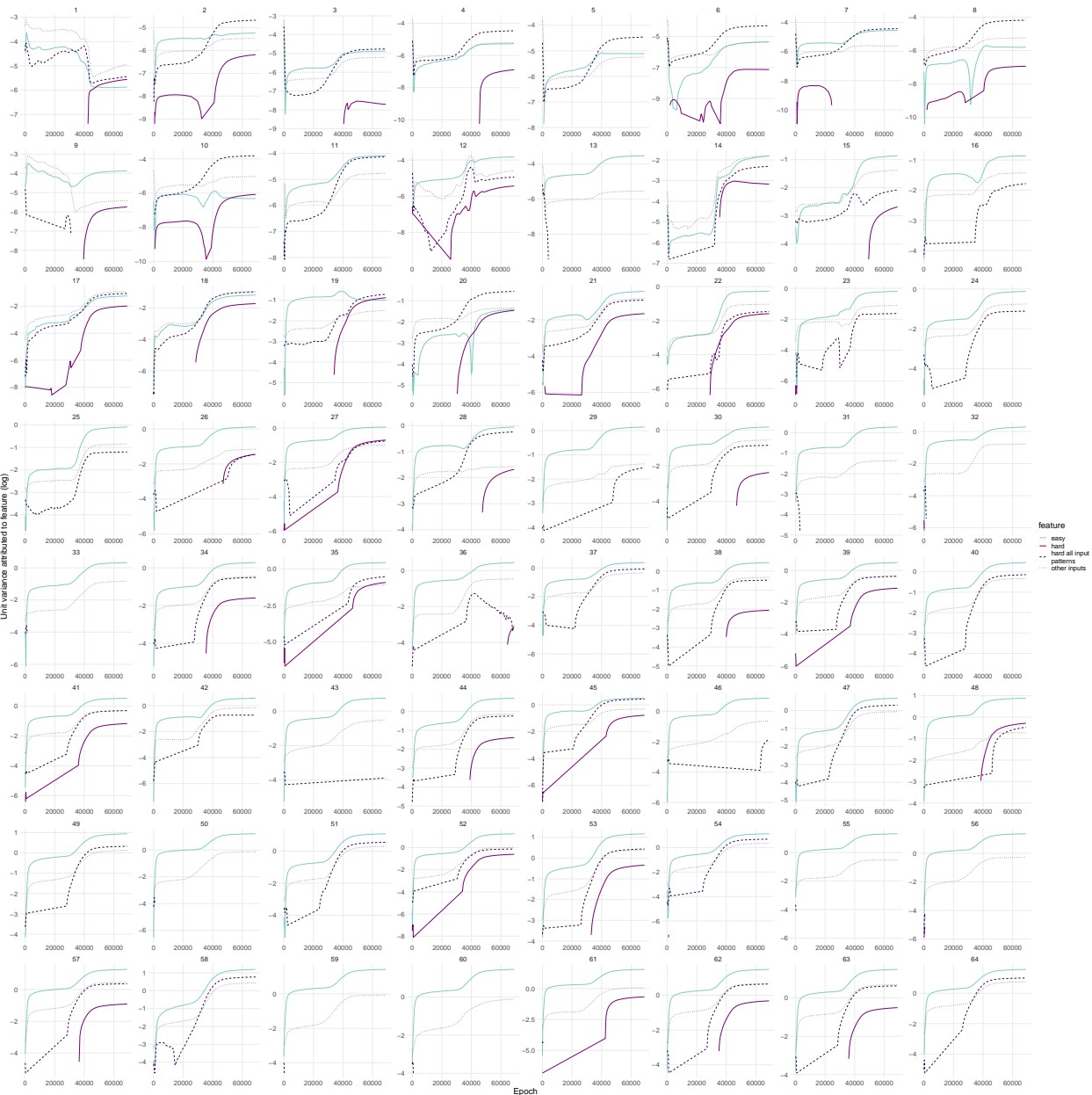

Figure 23: Raw representation variance plotted across learning for each of the 64 units in the penultimate layer of an MLP. Note that this figure plots raw variance explained, without normalizing by the total variance of the unit, to facilitate comparing across units. Note also that the vertical axis is log-scaled; where the variance explained on the test set was ≤ 0 it is omitted from the plot.

## B.5 Feature complexity also biases sparsity of representations

Figure 24: Harder features also yield sparser representations. (Lines are means across 10 seeds, errorbars are 95%-CIs).

We also find that more complex features tend to be represented more sparsely in the models representations (Fig. 24). To measure sparsity, we evaluate the hidden representations, and then standardize element-wise so that each unit has variance 1. We then fit a logistic regression decoding the feature, and evaluate the entropy of the softmax of the absolute value of the regression coefficients. If the regression integrates information more uniformly across all the units, this entropy value will be higher, while if fewer units are contributing, the entropy value will be lower; thus, we subtract this measure from the maximum entropy (a uniform distribution) and divide by the maximum entropy to calculate a 0-1 sparsity score for each feature. By this measure, the harder features are noticeably sparser in the models representations, even in the case that they are pretrained.

## B.6 More features of varying complexity

While our main MLP experiments only trained models to compute two features, deep networks trained on real tasks will compute many more. To move slightly closer to this regime, we trained networks to predict four features (N.B. we also use more features in our language and vision experiments), which we randomly sampled from a set of five features of varying complexity. We trained the models to predict all features simultaneously, and as above evaluated test accuracy and variance explained by each feature.

We show the results in Fig. 25. In general, the patterns are roughly what would be expected from the two-feature experiments above: adding more simple features to the dataset tends to suppress more complex features, while adding more complex features inflates simpler ones. Note that we cannot disentangle these effects fully here, but see the learning curves above Appx. B.3 for evidence that both inflation and suppression happen in the two feature case. Features of intermediate difficulty tend to inflate simpler features but interfere with more complex ones.

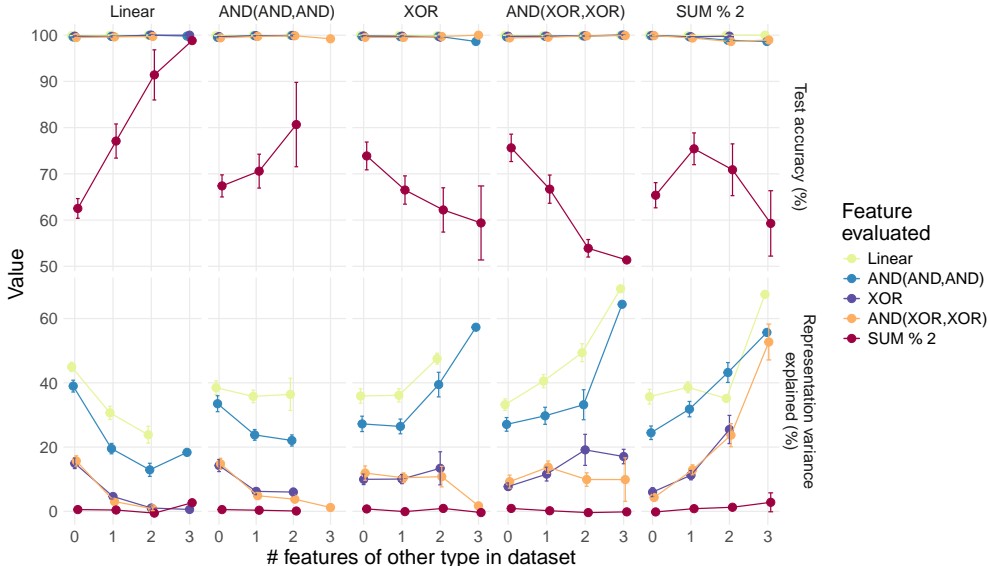

Figure 25: Training MLPs on four features, sampled from a set of varying complexity. This figure shows the effect on test accuracy (top) and representation variance explained (bottom) of a certain type of feature (colors) from the number of features of another type (panels). For example, the bottom left panel shows the representation variance of each feature type, as a function of of how many of the other features in the dataset are linear. As the number of linear features in a dataset increases, the variance explained by other features decreases noticeably. Conversely, the presence of more hard features in the dataset tends to inflate the variance of all simpler features. Features of intermediate difficulty, such as XOR, tend to inflate simpler features, but suppress equally or more complex ones. (Note that due to small dataset size, the hardest feature is *only* generalized reliably when all other features are linear. Results from training on 128 datasets with randomly sampled sets of four features. Lines are averages, errorbars are 95%-CIs.)

## B.7 The effects of MLP architecture and other hyperparameters

How specific are our findings to the particular MLP architecture we evaluated? In Fig. 26 we show the effect of depth/width/aspect ratios (i.e. ratios of layer sizes) of the MLP on our findings; in short, deeper MLPs show somewhat larger gaps, width and aspect ratio appear to have little effect. Residual MLPs also show qualitatively similar biases (Fig. 27), in contrast to our finding that ResNets have strikingly different biases to CNNs (§11).

Prior work has found that tanh and rectifier networks have qualitatively different representation-learning properties (Alleman et al., 2024), thus we performed a similar comparison here (Fig. 28). Overall patterns are similar, but we do note a slightly smaller gap in general with tanh networks. This observation may fit with the results of Alleman et al., who found that tanh network representations were generally more driven by the output structure, while rectifier network representations were driven more by input structure; that finding might suggest a somewhat smaller gap for tanh networks in our setting, since the output structure is identical for both features, but the input structure differs.

In the main experiments we encoded the easy feature redundantly across 4 input units (to match the number of inputs used for the hard feature); in Fig. 29 we show that this redundant encoding of the easy feature slightly increases the gap, but does not change the qualitative pattern of results.

In our main experiments, we used variance scaling initializers with an initialization scale of 1.0 for the network weights (that is, we initialized the weights to approximately preserve the variance of signals through the network at the beginning of training). However, we show in Fig. 30 that smaller initialization scales yield somewhat larger biases towards the easy features.

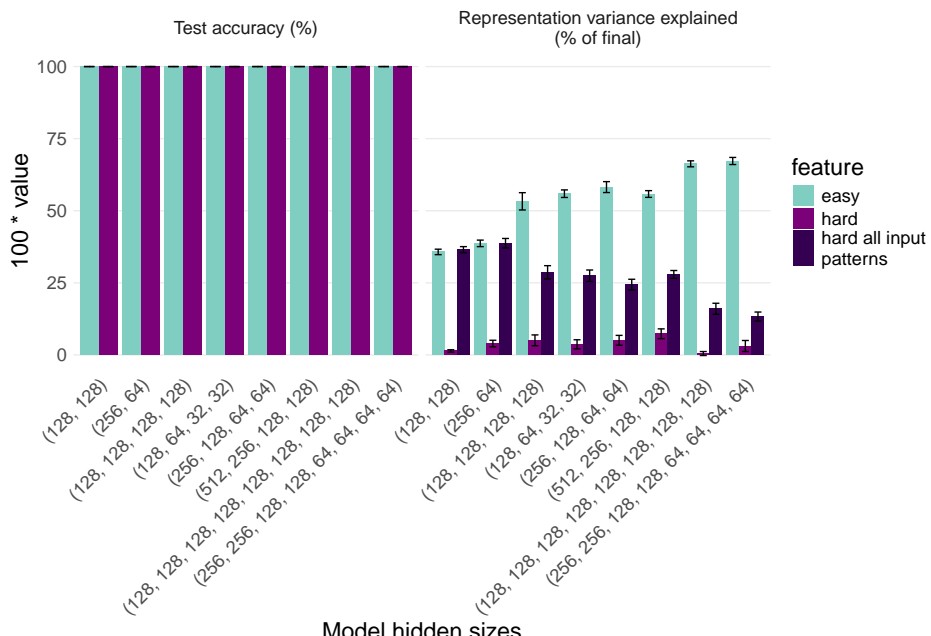

Figure 26: Exploring the effect of architecture on the gap between easy (linear) and hard (sum % 2) features. All architectures generalize well. Layer width does not seem to have a substantial impact on the representation gap in most cases. However, deeper models tend to show larger gaps. (Bars are means across 10 seeds, errorbars are 95%-CIs).

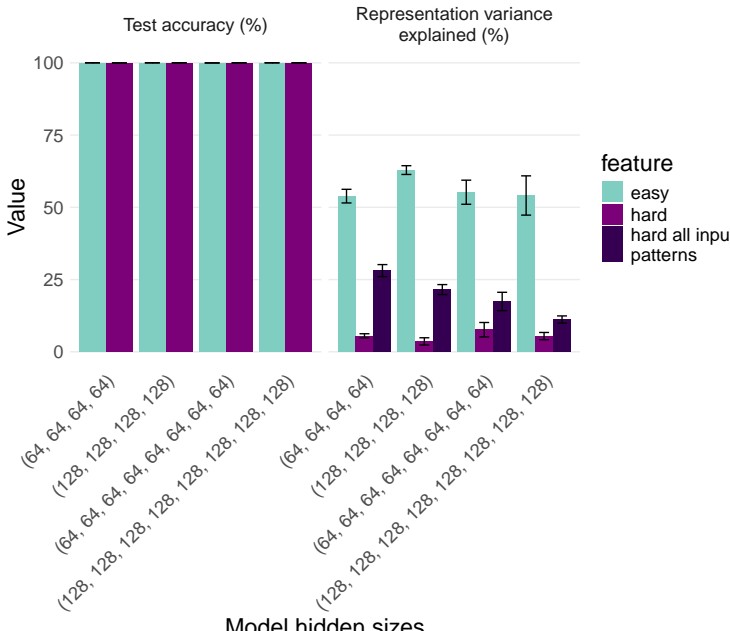

Figure 27: MLPs with residual connections also show qualitatively similar patterns of bias. (Bars are means across 10 seeds, errorbars are 95%-CIs).

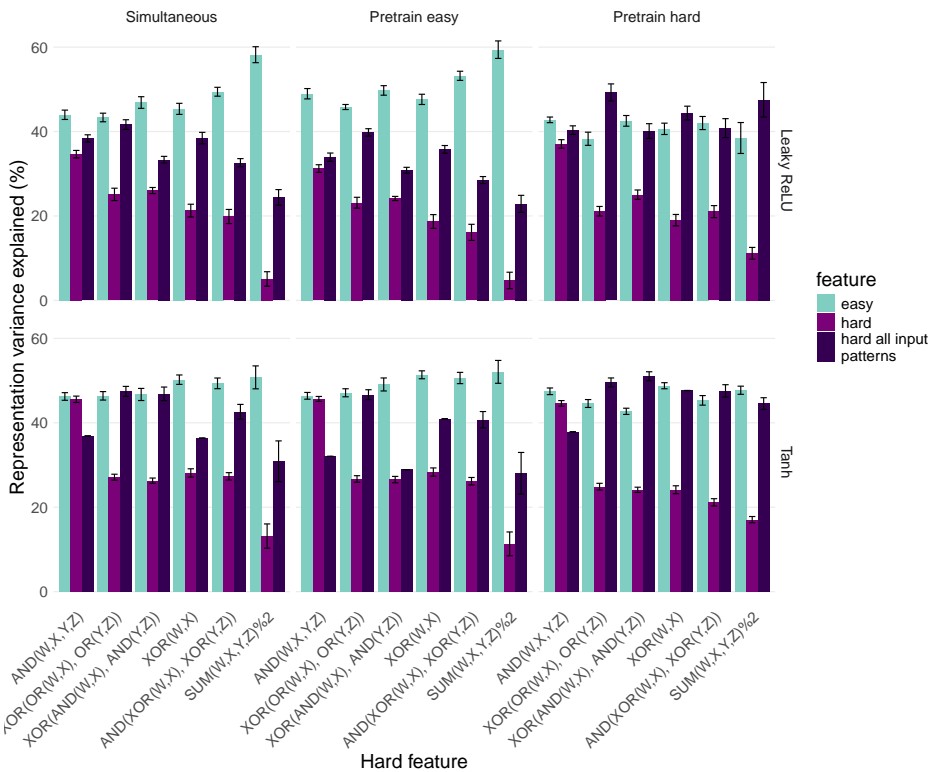

Figure 28: Effect of nonlinearity (rows) on the results. Patterns are qualitatively similar with either Leaky ReLU (as used in our other experiments) or tanh nonlinearities; however, tanh shows somewhat smaller gaps in general. (Bars are means across 10 seeds, errorbars are 95%-CIs).

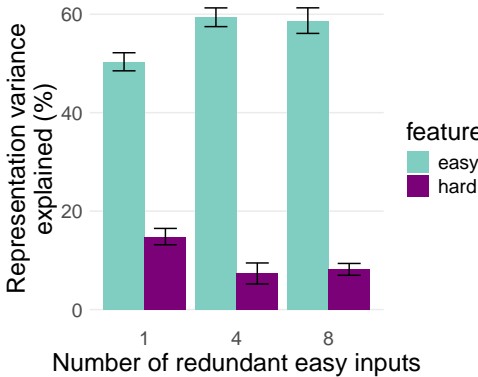

Figure 29: Redundantly encoding the easy feature across 4 or 8 input units results in slightly stronger gaps between easy and hard features, but qualitative results are similar with only a single easy input. (Bars are means across 5 seeds, errorbars are 95%-CIs).

### B.7.1 Dropout and optimizer interact for MLP architectures

The results we observed in the main experiments were with SGD. In Fig. 31, we show that various other optimizers (e.g. Kingma & Ba, 2014; Loshchilov & Hutter, 2017; Chen et al., 2024) show similar effects without dropout. However, in Fig. 32 we show that enabling dropout has different effects on different optimizers at convergence. For SGD and adagrad, the bias direction is consistent across dropout rates, though the bias magnitude shows a minimum at moderate dropout rates, but for other optimizers the biases reverse

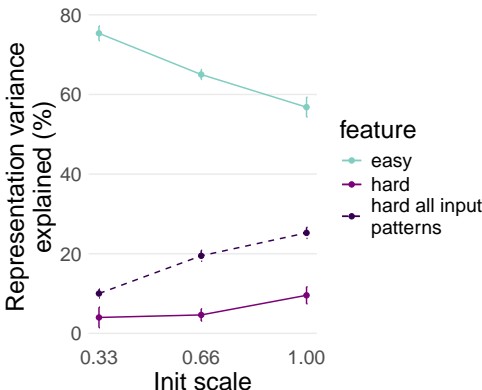

Figure 30: Smaller weight initializations yield somewhat larger biases towards easy features. (Bars are means across 5 seeds, errorbars are 95%-CIs).

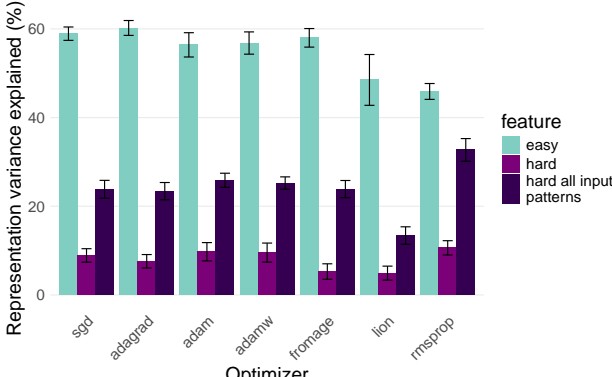

Figure 31: Effect of optimizer on the results without dropout. Patterns are qualitatively similar with all optimizers. However, we see more dramatic interactions with dropout enabled, see Fig. B.7.1. (Bars are means across 10 seeds, errorbars are 95%-CIs.)

or disappear at moderate or high dropout rates. However, this effect only occurs at convergence; earlier in learning the easy features dominate, even after the hard features are generalized fairly well. Furthermore, in Fig. 33 we show that the feature biases in Transformers trained on sequence datasets are more consistent, even with dropout enabled.

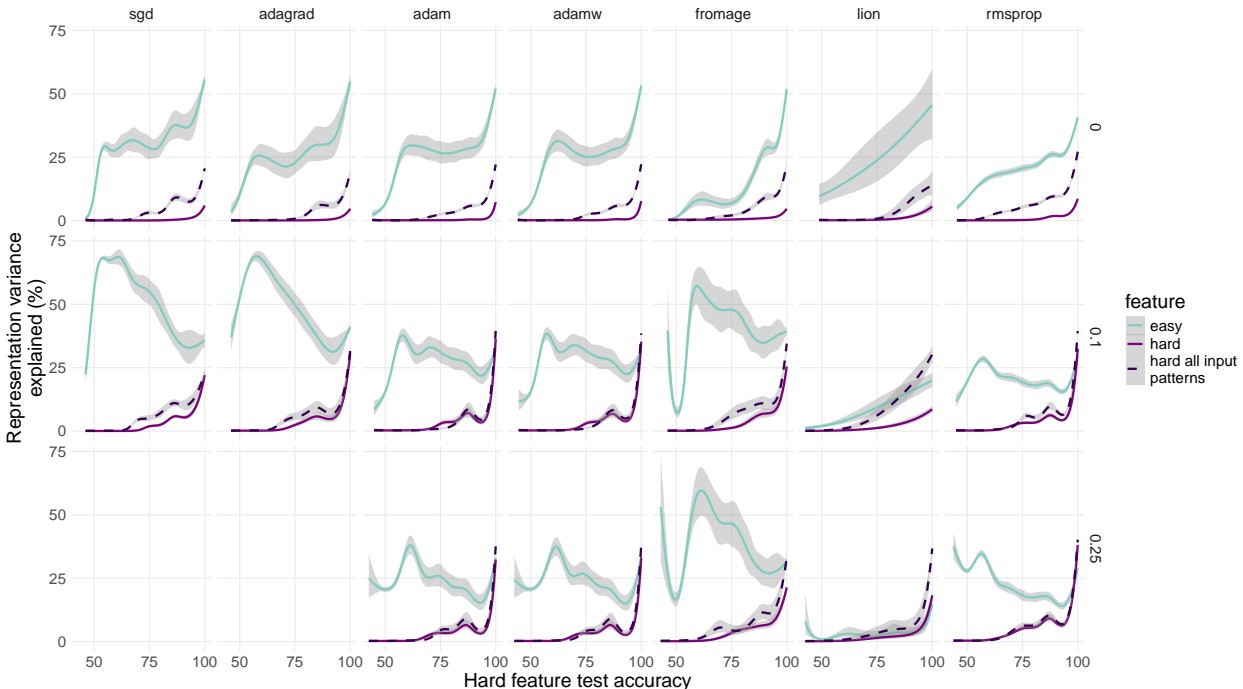

Figure 32: There is a strong interaction between dropout rate and optimizer for MLP architectures, but only at convergence. To visualize the results despite very different learning speeds with the different optimizers, this plot visualizes representation content as a function of hard feature learning, by plotting *hard feature test accuracy* on the horizontal axis. While all optimizers show similar patters of effects without dropout, and with dropout even at fairly late stages of learning the hard feature (90-95% generalization), dropout causes different effects at convergence with different optimizers. SGD, and Adagrad show consistent gaps between easy and hard features where they converge, but most other optimizers show bias towards the *harder* features at moderate or high dropout rates. (Bars are means across 10 seeds, errorbars are 95%-CIs. SGD and AdaGrad failed to learn with high dropout rates, so they are omitted from the last row.)

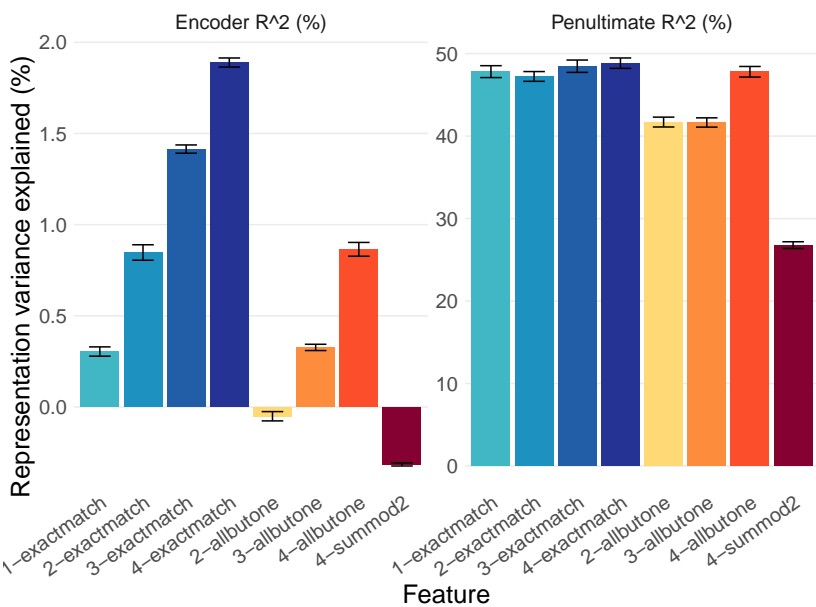

Figure 33: Unlike the MLP results above, the feature biases of Transformers trained on the more complex sequence tasks (cf. Fig. 7) remain fairly consistent even with dropout enabled. Note that this plot is produced at convergence, all models achieve >99% test accuracy. Thus, the bias switch at convergence seems less likely to occur in larger models trained on more complex tasks. (Bars are means across 64 randomly sampled datasets and seeds, errorbars are 95%-CIs.)

### B.8 Transformer sequential learning curves

In Fig. 34 we show learning curves for transformers trained on the letter-sequence tasks, with one feature enabled at a time. We observe some representation biases due to feature order, more so at the encoder layer. In fact, there seems to be relatively little change in the feature ranking at the encoder layer once the first few features are learned. One possible explanation is that the encodings may be "good enough" to not require substantial further learning to identify other features at other positions, thus leaving a lingering bias.

At the decoder layer, we see more complex patterns. We use 100 epochs to smoothly transition to adding the loss on the new feature, but even so we observe substantial interference, which increases with each subsequent feature. We also see some decay towards the end of learning each feature, likely due to the weight decay. The magnitude of bias appears to vary as different features are trained. Note that because the transformer's processing is shared across time points, variance explained for a given feature tends to increase even before its loss weight is enabled. The weight sharing may also contribute to the overall inflation of the variance explained by each feature as learning progresses (the staircase pattern), even once that feature has been completely learned.

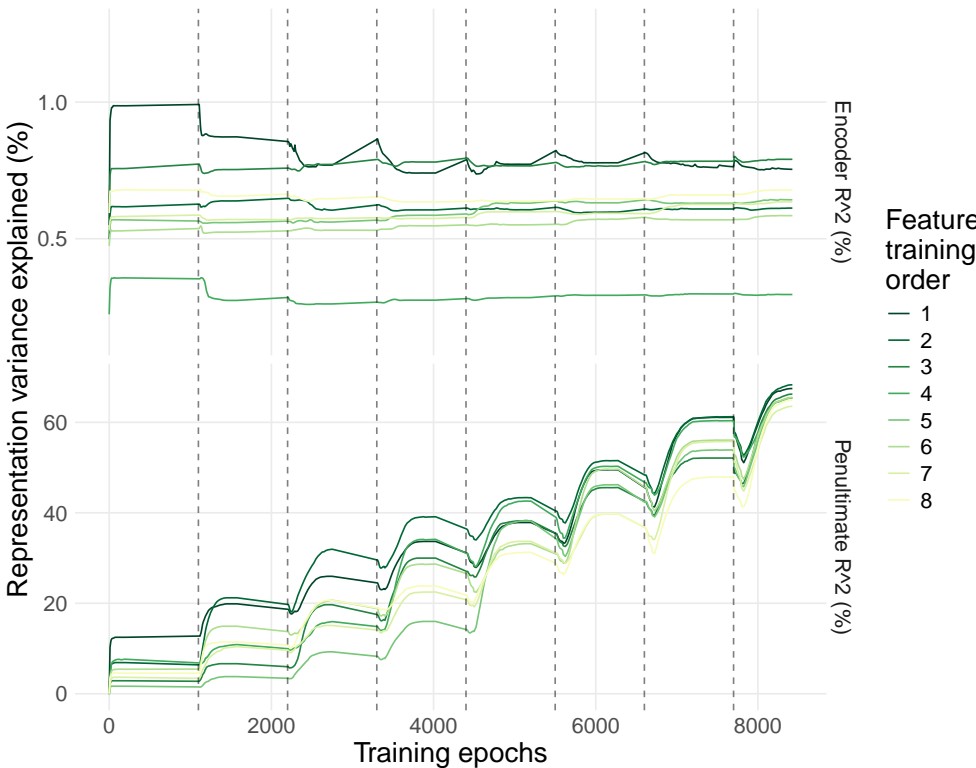

Figure 34: Learning curves for sequential training in Transformers. Feature order biases are noticeable, but appear weaker than in the MLPs, at least at the penultimate layer. However, several other qualitatively interesting patterns appear, including very little change in the encoder after the first few features are learned, and decay, interference, and recovery in the decoder. (Lines are averages across 48 seeds. Dashed lines indicate when the loss begins to ramp up for each feature.)

### B.9 Additional Dyck language results

In this section we present some further analyses of the transformers trained on the Dyck languages. First, in Fig. 35 we present the representation variance explained by each feature without controlling for sequence length.

Second, as mentioned in the main text, the models do not learn these features equally well (Fig. 36). Thus, in Fig. 37 we plot the variance explained by each feature as a function of test accuracy. We do not see a clear correlation between how well a feature is generalized and how much variance it explains, (although the variability in each feature's test accuracy is low, which might reduce our power to detect such an effect).

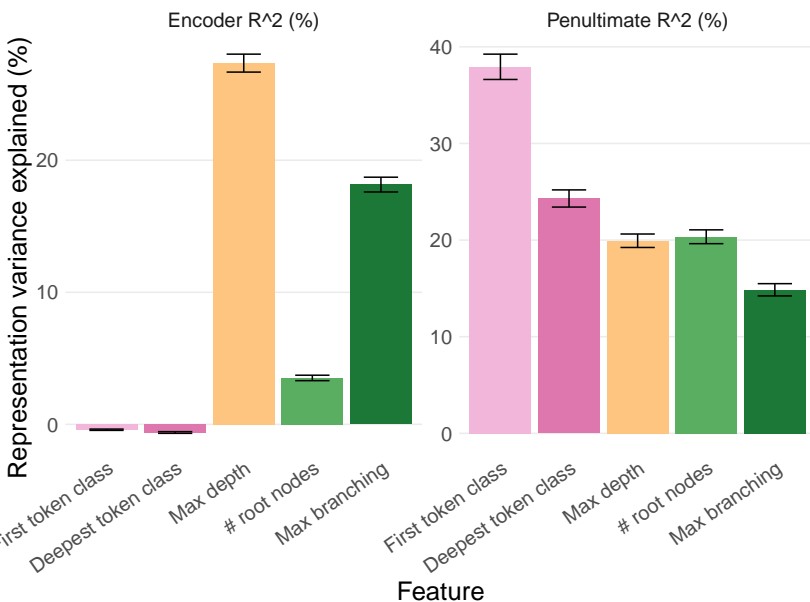

Figure 35: Representation variance explained without controlling for length at the final encoder layer (left), and pre-logits decoder layer (right) for transformers trained to identify semantic and structural features of sentences from Dyck-(20,10). Results are broadly qualitatively similar to the length-controlled results in Fig. 9, but the correlations with length change the numerical values. (Bars are averages across 32 seeds, intervals are 95%-CIs.)

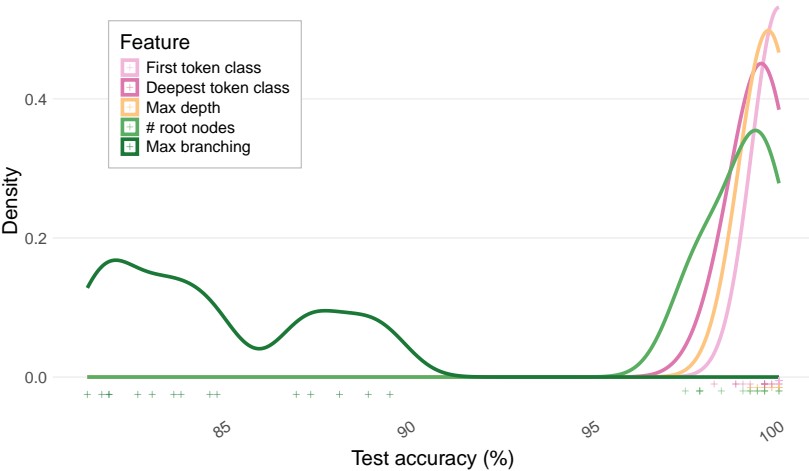

Figure 36: Test accuracy distributions (across runs) on superficial and structural features of sentences from Dyck-(20,10).

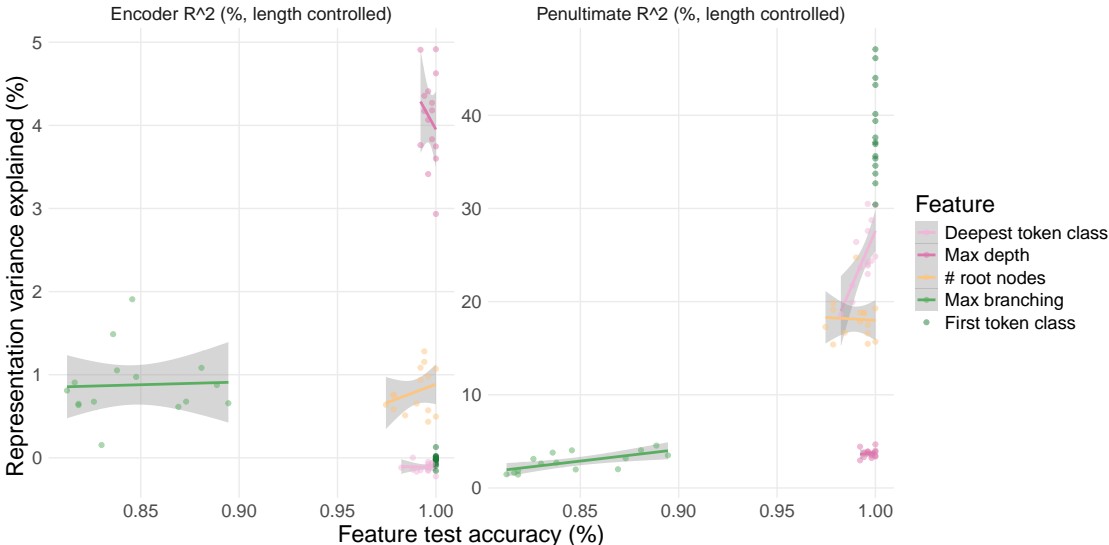

Figure 37: Representation variance explained at the final encoder layer (left), and pre-logits decoder layer (right) vs. test accuracy for transformers trained to identify superficial and structural features of sentences from Dyck-(20,10). (Lines are linear model fits across 32 seeds, intervals are SE.)

## B.10 Additional vision experiment results

### B.10.1 ResNet representational biases for shape are affected by the number of class labels and the presence of distractors

We noted a surprising shift in representational biases of ResNets between our two vision settings: favoring shape in one case, but favoring color and texture in another. Here, we disentangle the contributors to this effect, by removing some of the confounding changes.

First, in Fig. 39a we show that these changes are not driven solely by the shift to binary labels; if we simply replace the original single-object classification task with the binary-label version, without changing the input distribution or adding new objects, we still see shape biases (though they are somewhat less pronounced).

Second, we keep training only on binary classifications of the basic features, but switch to the *input* distribution of the relational tasks; that is, we effectively introduce two distractor objects, which the model simply has to ignore (because we train it only to report binary classifications of the first object). Surprisingly, the biases shift in this case to favor color and texture over shape (Fig. 39b).

Thus, because the biases were reduced with binary labels in the original setting, and flipped in the multi-object setting, it seems that both changes may contribute to the bias shift, but with a larger contribution from the distractor objects.

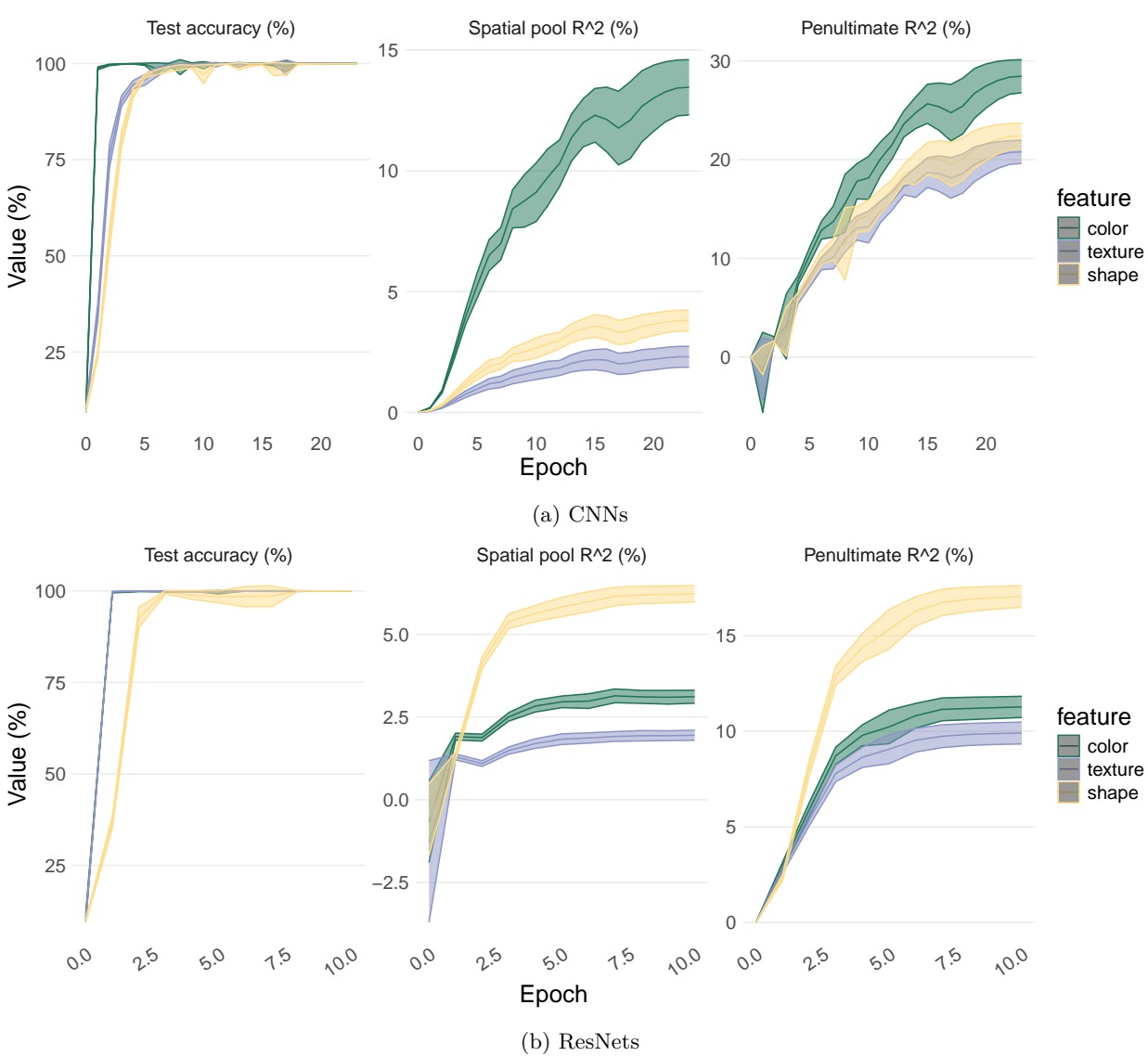

(a) CNNs

(b) ResNets

Figure 38: Learning curves for test accuracy and variance explained for CNNs and ResNets trained on the color-texture-shape tasks. (a) For CNNs Color, which is learned first, quickly comes to dominate at the spatial pooling layer; the penultimate layer produces relatively less bias. (b) For Resnets, color briefly dominates, but is quickly overtaken by shape. (Lines are averages across 32 seeds, areas are 95%-CIs.)

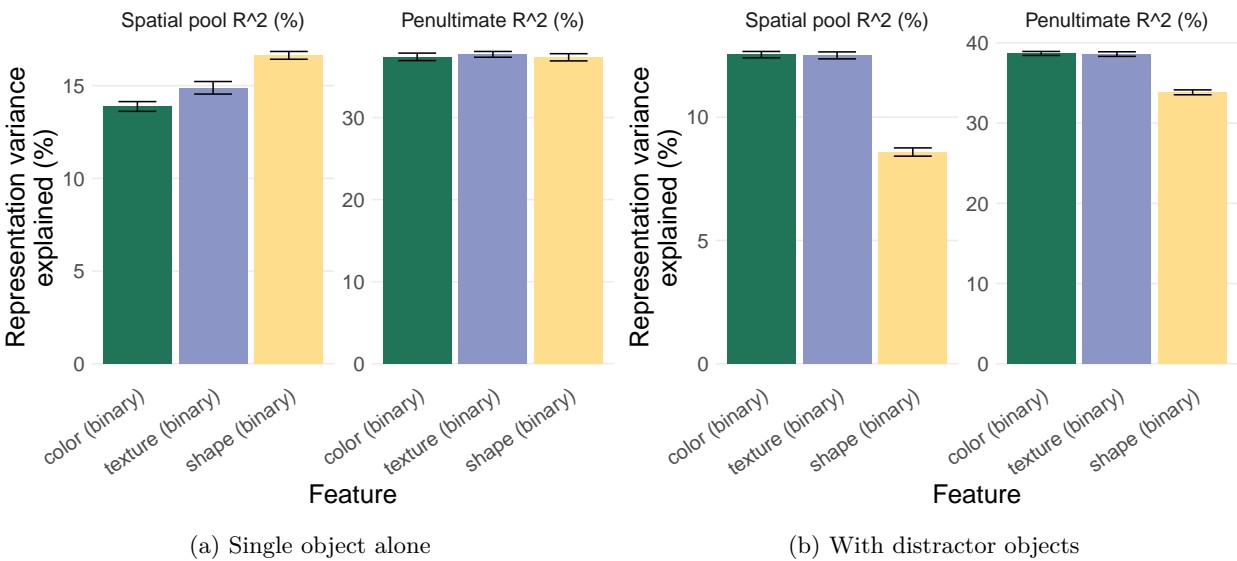

(a) Single object alone

(b) With distractor objects

Figure 39: Representational biases of ResNets trained to report binary classifications of the color, shape, and texture of a single object shift depending on the other objects in the image. (a) with no other objects in the image, the representations still show shape bias, but more weakly than with the original 10-way classifications (Fig. 11b). (b) With distractor objects in the image, the biases shift to favor color and texture. (Lines are averages across 32 seeds, areas are 95%-CIs.)

### B.11 Keeping all but the top-$k$ principal components

In Fig. 40 we show that if we *drop* the top principal components from the model's representations (rather than preserving them, as is typical), the predictions of the hard feature are robust to dropping more components, because the top components are occupied with the easy feature.

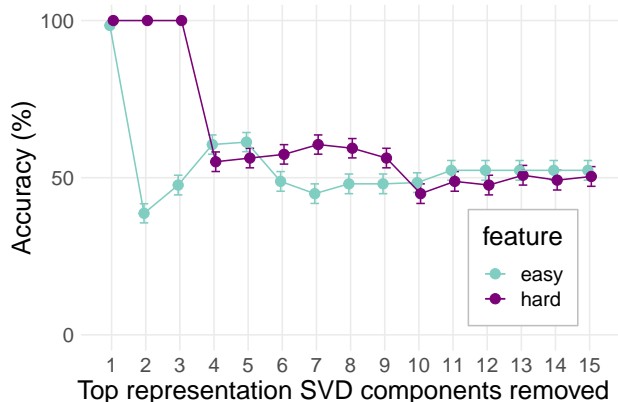

Figure 40: If we *drop* the top-$k$ principal components from a models representations, the easy feature predictions will be harmed at a smaller $k$ than the hard feature. (Cf. Fig. 14.)

### B.12 Low-dimensional visualizations

In this section we show several low-dimensional visualizations of the representation space of models trained on a complex feature (sum % 2) and a varying number of easy features (1-4).

In Fig. 41 we show visualizations of the first six principal components of the models. The easy features always occupy at least the first two principal components. In the case that there is only a single feature, the hard feature is separated along the third principal component. As more easy features are added, they occupy successively more principal components, and the harder feature is correspondingly shifted one PC higher for each easy feature that's added. Given this, it is somewhat surprising that *two* PCs are devoted to the single easy feature in the one easy-feature case, when just one would suffice.

In Fig. 42 we show *t*-SNE visualizations of the same networks. It is noticeably easier to see the hard feature separating the clusters in these plots than in the top few principal components, which demonstrates the benefits of a nonlinearity and locality bias of *t*-SNE. Nevertheless, as more easy features are added, it becomes hard to tell that there is another feature separating the clusters, particularly in the case of 4 easy features.

Taking a step back, how would the results of this section extrapolate to more realistic settings? If we imagine a more realistic problem regime in which tens, hundreds, or perhaps even thousands of features contribute to a model's predictions, it seems quite likely that some of the computationally-important-but-complex features would be entirely swamped by the noise of the easier features in any typical feature visualization.

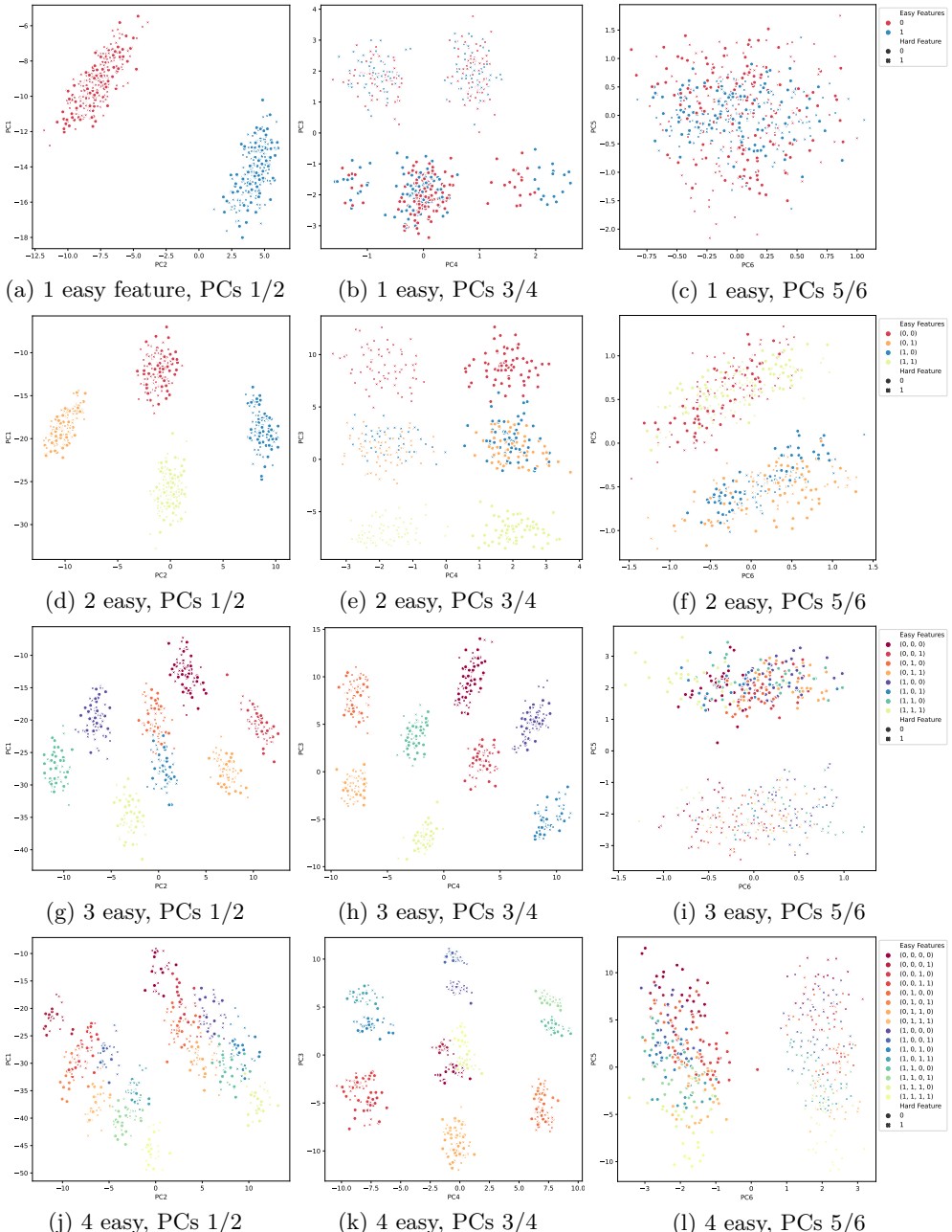

(a) 1 easy feature, PCs 1/2      (b) 1 easy, PCs 3/4      (c) 1 easy, PCs 5/6

(d) 2 easy, PCs 1/2      (e) 2 easy, PCs 3/4      (f) 2 easy, PCs 5/6

(g) 3 easy, PCs 1/2      (h) 3 easy, PCs 3/4      (i) 3 easy, PCs 5/6

(j) 4 easy, PCs 1/2      (k) 4 easy, PCs 3/4      (l) 4 easy, PCs 5/6

Figure 41: Principal components visualizations of penultimate representations from MLPs that are trained to output a fixed hard feature (sum % 2), together with different numbers of easy linear features (increasing from 1-4 across rows). As easy features are added, their labels (colors) spread across more principal components (columns), and thus the dimension along which the hard features (shapes) separate is pushed into higher and higher principal components. With 1 easy feature (top row), the first two PCs are clustered according to the easy feature (a), but the hard feature is clustered along the third principal component (b). When two easy features are learned, the third principal component is used to help represent them, and the hard feature is pushed into the fourth principal component (e). When three or four easy features are learned, the hard feature is pushed into the 5th or 6th PCs, respectively (i & l). There is a remarkably consistent progression that as each easy feature is added, another PC is captured, and the hard feature is pushed one PC higher.

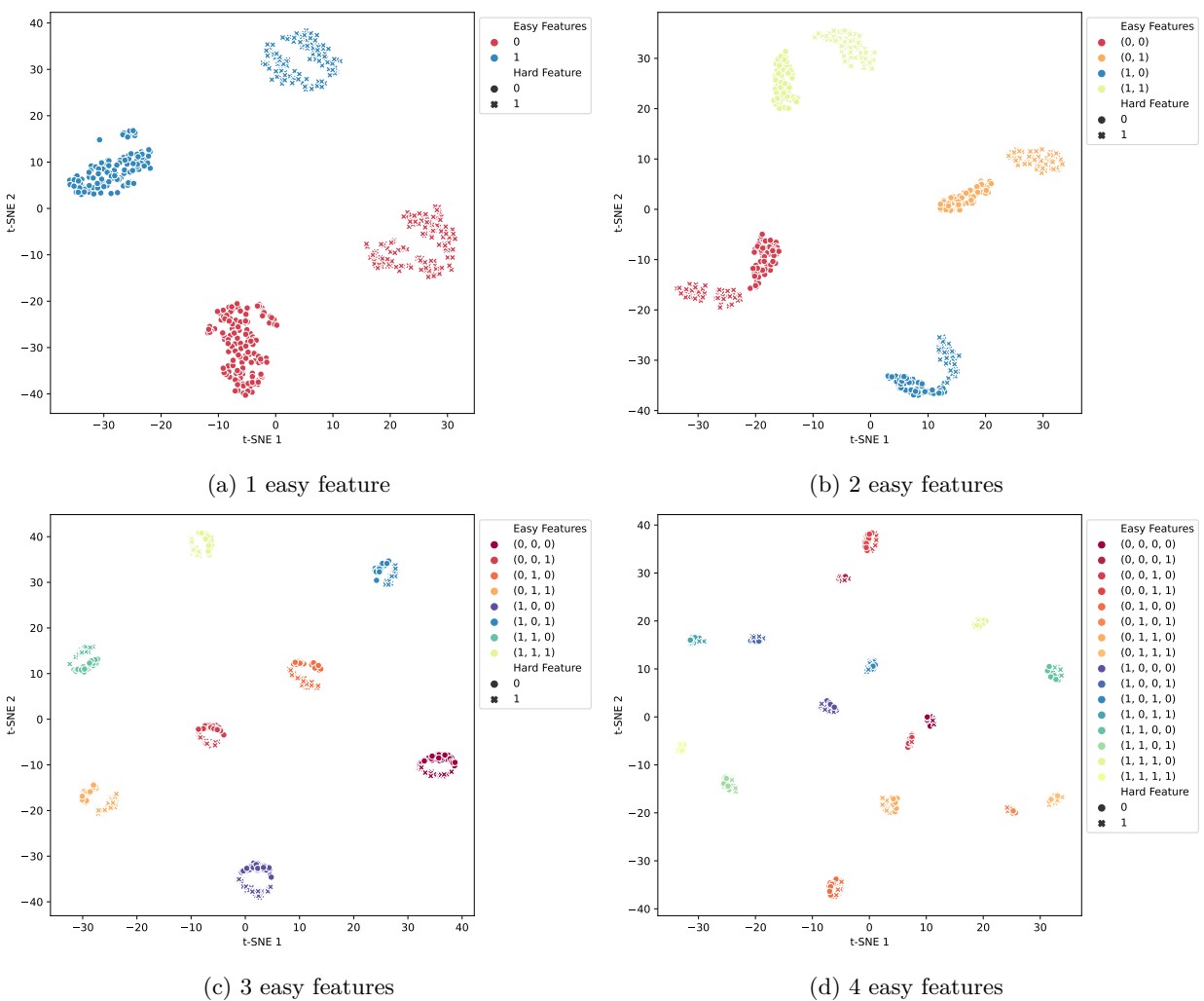

(a) 1 easy feature

(b) 2 easy features

(c) 3 easy features

(d) 4 easy features

Figure 42: $t$-SNE visualization of the penultimate representations from MLPs that are trained to output a fixed hard feature (sum % 2), together with different numbers of easy linear features (increasing from 1-4 across panels). With a small number of easy features, the clustering according to both the easy feature values (colors) and the hard features (shapes) emerges quite clearly. The existence of the hard feature is generally more readily visible in these $t$-SNE plots than in examining just the top few principal components (cf. Fig. 41). However, as more easy features are introduced, they come to dominate the representation space, and with 4 easy features it would be difficult to discriminate the hard features separating most clusters if they were not already labelled with shapes.

### B.13 RSA result resilience

In Fig. 43 we show that the RSA results in the main text are qualitatively similar if different metrics are used for the analysis, or if Spearman rank correlation is used to compare the dissimilarity matrices.

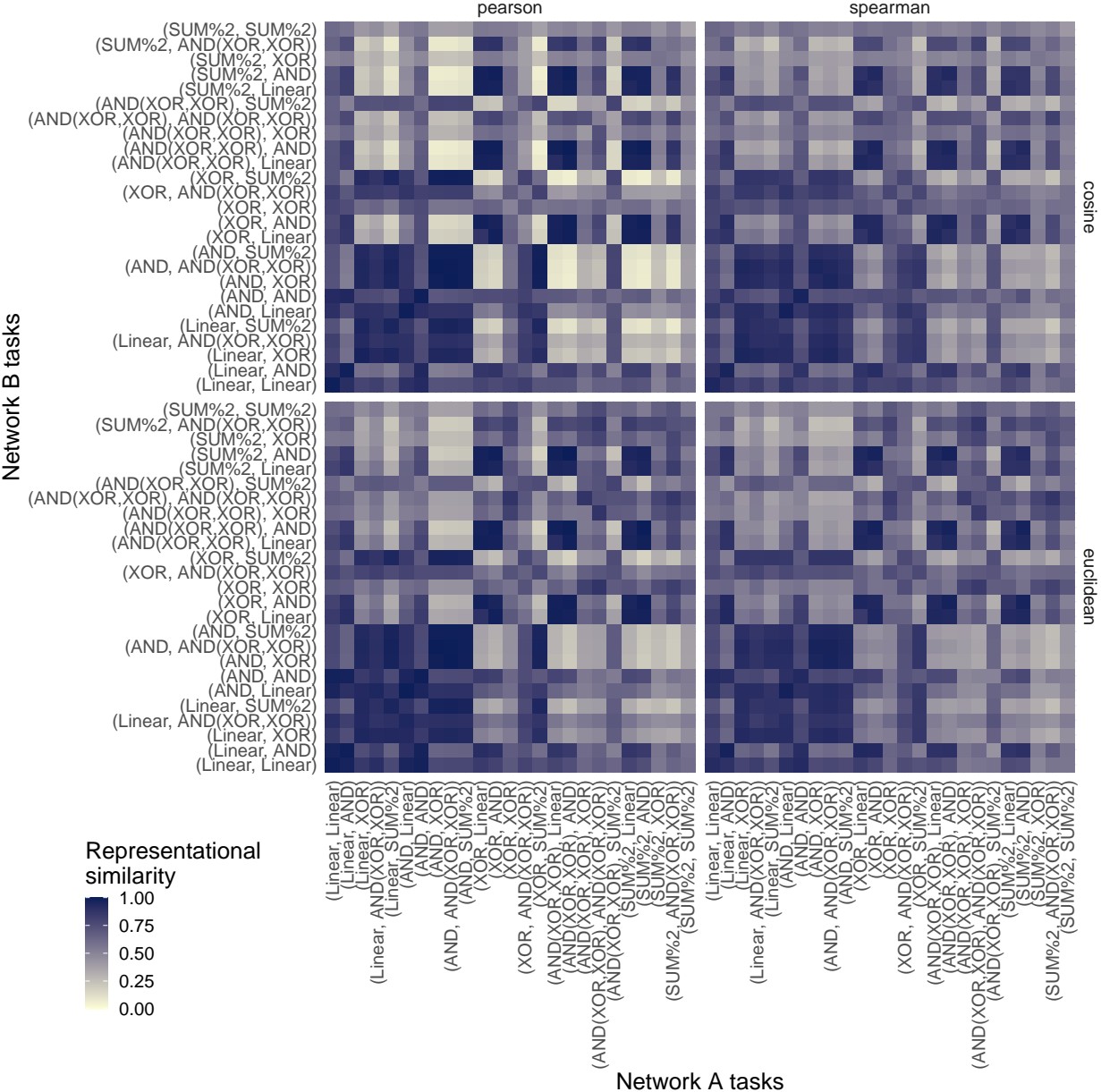

Figure 43: RSA results with different dissimilarity metrics (Cosine distance on the top, Euclidean distance on the bottom), and comparison correlations (Pearson on the left vs. Spearman on the right). Results are qualitatively similar in all cases. (Results are computed across 5 seeds per feature pair.)

