# OpenReview forum: "Learned feature representations are biased by complexity, learning order, position, and more"
_TMLR — Accepted by TMLR_

### Review · Reviewer_VVSP · 2024-07-22

**Summary Of Contributions:**

This paper empirically finds that learned feature representations are biased by various factors, such as feature complexity, the learning order of different features, and the distribution of features. Experiments show that these biases are affected by architectures (i.e., MLPs, Transformers and ConvNets) and training regimes.

**Audience:**

Yes

**Broader Impact Concerns:**

No concerns.

**Claims And Evidence:**

Yes

**Requested Changes:**

See weaknesses.

**Strengths And Weaknesses:**

**Strengths:**

This paper investigates representation bias in deep learning models by systematically manipulating feature attributes and the training regimes, which can potentially offer insights into how deep models work.

**Weaknesses:**

1.	The experiments focus on toy/synthetic datasets, which may limit the generalizability of conclusions about representation bias to more complex tasks. For example, in vision tasks, the authors suggest that "features like color are easier for models to learn." Does this conclusion hold across different model architectures, such as classic ConvNets (AlexNet, ResNet, VGG, DenseNet, etc.) and Transformer-based architectures (ViTs, Swin Transformers)? Furthermore, how do these findings extend to more complex tasks? Therefore, the authors need to consider the applicability of their conclusions about representation bias, e.g., whether they are task-specific, architecture-specific, or both. Additionally, validating these findings on more complex real-world datasets and across a wider range of model architectures would provide a more comprehensive understanding.

2.	The paper needs to provide clear mathematical definitions for important concepts, instead of relying on vague textual descriptions. For instance, in Section 2, the authors mention "...even when the features play similar computational roles". What are "similar computational roles"? Does this refer to the similar output vectors or probabilities? To address this, the authors should clarify the important concepts in Section 2, such as "input stimulus", "feature", "feature representation", and "representation variance explained by a feature", providing precise mathematical definitions or formulas. If it is challenging to abstract these concepts into unified mathematical definitions across different tasks, the authors could provide specific definitions for classic tasks.

3.	For visual tasks, the authors should provide examples of the synthetic datasets used. Furthermore, the authors should specify the details of the non-residual CNNs, including the number of layers, the network structure, and whether all non-residual CNNs, such as VGG and AlexNet, are considered.

4.	Minor: There are problems with the layout of the figures. For example, the caption of Figure 9 is truncated and cannot be fully displayed.

---

> ### Author Response · Authors · 2024-07-26
> **Response**
>
> We thank the reviewer for their insightful questions. We attempt to address their concerns below:
>
> 1. We have provided some added discussion of the limitations in the current work (see common response) as well as referencing a paper that was released after ours was submitted that explores some related issues in a less controlled setting, but with larger models, by studying feature learning in ImageNet (https://arxiv.org/abs/2407.06076).
> 2. We agree that mathematical definitions help to clarify our methods and analyses — we have added more formal definitions for some concepts (e.g. features) and revised the formal definitions for the feature variance measures in light of them; we have also emphasized the formal definitions in the methods further.
> 3. We thank the reviewer for pointing this out — we had included example images and the architecture details in the appendices, but had failed to reference them from the main text so they were difficult to locate. We have now added those references.
> 4. We made a minor change before submitting that moved many images and messed up the formatting; this has been fixed in the revised version.
>
> We hope these changes will address the reviewer's concerns.

---

### Review · Reviewer_qC5t · 2024-07-22

**Summary Of Contributions:**

The paper explores the biases of learned feature representations in neural networks. By feature they mean a predicted property of the raw input, "Does the sequence contain 'abc'?" or "What is the color?". Representation is some encoded form of the input which is typically the activations of the last layer of a neural network.

They come up with a systematic way to explore how much variance is explained in the representations due to a specific feature. The feature may be "easy" or "hard" which is loosely defined as how many nonlinear transformations of the input is needed to compute the feature. They carry out experiments with synthetic data and several different model architectures.

They consistently find that most of the variance is explained by these simpler features. Things like learning order or residual connects seem to affect the affect the bias. They discuss the implications this has for mechanistic interpretability work that uses Principal Component Analysis (PCA) or really anything that uses a metric similar to L2 reconstruction loss since the work would suffer from the same bias.

**Audience:**

Yes

**Claims And Evidence:**

Yes

**Requested Changes:**

The paper is good enough as is, but I would be curious to see more work on larger models and a better understanding on why residual nets seem to have different biases.

**Strengths And Weaknesses:**

The paper flows well using the simple MLP setting and then generalizing to transformers to CNNs. They accompanying notebook is simple and makes things clear, too. They do a good job of contextualizing the work with transformers and dictionary learning with sparse autoencoders.

Some of these methods may behave differently with much larger models that are much deeper. I feel like the different behavior or residual nets is under-explored.

---

> ### Author Response · Authors · 2024-07-26
> **Response**
>
> We thank the reviewer for their comments, and are glad they found the work to be useful and clear. We agree that it would be useful to explore deeper models and more realistic settings, and have included some further discussion of this in an expanded limitations section (see common response) as well as referencing a work that was released after our paper was submitted (https://arxiv.org/abs/2407.06076), that explores some related issues in an ImageNet-trained vision model. We believe that future work should explore these issues more deeply across a broader range of more realistic settings.

---

### Review · Reviewer_GLvA · 2024-07-23

**Summary Of Contributions:**

The authors train neural networks to predict features of the input. They then examine properties of how these features are learnt by varying properties of the setup, such as feature complexity, learning order, and the distribution of features.

**Audience:**

Yes

**Broader Impact Concerns:**

I do not think there are any real ethical concerns with this paper.

**Claims And Evidence:**

Yes

**Requested Changes:**

As it currently stands, I do not think the work is sufficiently rigorous for publication in TMLR. As stated above, I am concerned by the fact that the authors seem to assume that features are embedded linearly and rely on this assumption throughout the rest of the article. In addition, many of the claims seem too far-reaching for the level of evidence provided.

I would like to see:
1. Focus on building experiments with much greater rigour. If the authors are going to analyse properties of feature learning assuming linear features, I would like them to study a model where we are much more confident that these features do end up being linear. In my eyes, the work would have been much stronger if the authors had simply focused on a very small number of training setups such as [othello-gpt](https://www.neelnanda.io/mechanistic-interpretability/othello) rather than many cases where they have not demonstrated that information is actually encoded linearly.
2. An improvement in the formatting and writing. Although TMLR is more accepting of papers without headline claims and more verbose discussion of results, I believe that the current writing needs to be much more succinct and structured. As it currently stands, when I read the paper I feel like I am gaining some interesting observations but no clear take-home messages.
3. A narrowing of claims made based on algorithmic datasets and a more rigorous handling of the differences between algorithmic and more natural datasets.

**Strengths And Weaknesses:**

I think the primary strengths of the paper are as follows:
1. I appreciate that the authors have provided their definitions for terms at the start of the paper. Given that there is no consensus answer to the question "what is a feature?", I think this is important.
2. The questions being asked are of interest to the community, in the sense that finding generalisable biases in feature learning order or dominance would be valuable.
3. I like the number of setups (although I think the number has limited the rigour).

Some more minor strengths:
1. I found the insight provided in section 6.2 noteworthy: "In particular, if the features required to solve the harder OOD splits are more complex, they may occupy relatively less variance, and therefore be more readily lost when keeping only the top principal components."

I think the main weaknesses are:
1. It seems that the way "variance explained" is measured is by fitting a linear predictor. This seemingly assumes a linear representation of the feature, which might be problematic if more complex features are represented non-linearly in the network. As such, the linear probe might explain less variance for complex features. This weakness may be somewhat ameliorated if the probes are trained only on the penultimate layer (especially if the last layer is an unembed rather than an MLP). However, it is still a major concern in my mind.
2. In section 3.2, how did the authors come up with "all patterns of inputs relevant to the hard feature"? What if the network is using a really unconventional representation that isn't included in this list of patterns? See [Progress measures for grokking via mechanistic interpretability](https://arxiv.org/abs/2301.05217).
3. Given that the tasks in section 3 and many of the tasks in section 4 are algorithmic, I'm unsure how well the results in those sections generalise. We know from grokking that the training dynamics and representations associated with algorithmic tasks may be very different from natural datasets. Given this, I think sentences like the following are claiming too much: "Thus, while these results show that the biases can be shifted in nontrivial ways by hyperparameter choices, we expect that easy feature biases will persist in standard training settings, where models are generally not trained to complete convergence on their tasks."

Some more minor weaknesses:
1. Could the authors please fix the formatting of the paper? For example, can page 8 and 11 be fixed?
2. I think that grokking and algorithmic tasks should be discussed more in the related work. I can provide some suggested literature if the authors would like.

I also have some questions:
1. Can you describe in more detail what was done in section 3.2? The weakness I have listed in this section may be a result of my misunderstanding.
2. Can the authors also provide further justification for the use of the linear predictor?

Update: after author changes, I have decided to change to "Yes" for "Claims and Evidence".

---

> ### Author Response · Authors · 2024-07-26
> **Response**
>
> We thank the reviewer for their thorough review and thoughtful concerns. We have tried to address these concerns where possible, and to note them in the expanded limitations section where not.
>
> * It seems that the primary concern of the reviewer was around our use of linear analyses. We have attempted to address this concern in two ways:
>    -  “Can you describe in more detail what was done in section 3.2?” — we have expanded the discussion in this section to try to clarify the process, and to elaborate on the implications. In brief, we enumerated all patterns of inputs relevant to the hard-feature inputs, and then denoted each pattern with a one-hot code. From this expanded one-hot space, **any** function of the hard inputs is achievable by a linear transformation (it is essentially creating a “tabular-like” representation in which the output for each input pattern can be expressed independently) — including but not limited to DFTs etc. Thus, this analysis should at least partially address the concern about basing our analyses on linear regression; from this augmented feature space we can detect nonlinear representations of the hard feature. This is still limited, in the sense that it cannot accommodate arbitrary nonlinear functions of the hard inputs together with other inputs, but enumerating the full space of nonlinear representations is infeasible without much larger datasets than we worked with (and seems beyond the scope of this work unless there are clear motivations for doing so). Thus, we believe this analysis should address the reviewer’s concern.
>     - We have also included some further discussion of our use of linear predictors in the new limitations subsection (see shared response), including discussing their extensive use in the prior literature, which we believe makes their assumptions reasonable vis-a-vis commenting on challenges of interpretability.
> * “I would like them to study a model where we are much more confident that these features do end up being linear. In my eyes, the work would have been much stronger if the authors had simply focused on a very small number of training setups such as othello-gpt…”: we admire the work that has been performed in Othello, but have several responses to this suggestion:
>     - First, we believe that in the MLP settings we provide as strong of evidence as the Othello work that the features are linear; viz. we analyze with linear regression and show that by intervening on the linear features we can precisely manipulate the model outputs (section 6.1).
>     - Furthermore, however, we believe that the Othello work **does not** provide strong evidence that the models are using *only* linear representations, as this comment seems to suggest. For example, there may be nonlinear features that are playing a role in the model computations, but that get causally preempted by interventions on the linear features in the analyses (cf. https://arxiv.org/abs/2407.04690v1). Thus, simply observing that there are linear features in the model in that work, and that interventions change model outputs, are not sufficient to claim the features “do end up being linear” overall. We have included some discussion of these latter issues in the new limitations section for our work.
>     - Finally, a key goal of our work is to focus on settings where we know the computational role of the features, and to make them statistically independent to avoid confounds; since in Othello or other tasks we do not know what the optimal solutions are, this would result in the kind of entangling of multiple features that we aimed to avoid in this work.
> * We have also included additional literature on algorithmic tasks and grokking, as well as fixing the formatting — see the common response for further details.
> * We have tried to improve the clarity and readability of the paper by providing takeaways as bullet points at the end of the introduction, and revising subsection titles to highlight the main conclusion from each.
>
> We hope these responses and changes will address the reviewer's concerns.

---

> > ### Comment · Reviewer_GLvA · 2024-08-04
> > **Response to authors**
> >
> > I'd like to thank the authors for taking seriously the concerns I expressed in my review and apologise for my late response. I like the steps that have been taken toward addressing my comments.
> >
> > In response to some of the points mentioned in the rebuttal:
> > * *Additional explanation of section 3.2.* As I now understand it, the relevant inputs is a list of length $2^5 = 32$ corresponding to the first $5$ input dimensions? I can now see that these would be useful for the model to express and thus useful candidate representations to look for and analyse. With this updated view of the process, I am less concerned about this section. I would, however, like to see the reasoning regarding the limitations of the approach mentioned in the paper when the approach introduced. Perhaps the authors could include a footnote mirroring the statement in their rebuttal that, "This [approach] is still limited, in the sense that it cannot accommodate arbitrary nonlinear functions of the hard inputs together with other inputs."
> > * I like the fact that further discussion of the use of linear predictors is included although I still have issues with it.
> > * Concerning othello-gpt. On reflection, I believe othello-gpt was likely a bad example to use to convey my point. On more detailed reading, I like the work done in section 6.1. What I would like is more verification in the style of section 6.1 rather than more case studies.
> >
> > In the time since my first review I have also had more time to think about the paper and have developed a few more concerns.
> > * The work only considers binary features. This is okay but I don't think this limitation is discussed enough. Features in networks trained on real-world data might have a strength encoded alongside existence. For example the magnitude of a linear vector encoding brightness might have information about the intensity of the light. I.e. $0.2 v_\{\text{brightness}}$ would represent greater brightness than $0.5 v_\{\text{brightness}}$. This is not captured in the setup used.
> > * Related to the above, I am concerned about whether the training setup used is really transferable outside of the context of the paper. I could not find in the paper what loss function is used for training the models analysed however this factor could have a large impact on the results. This is why, in my original review, I was advocating for analysing representations in a more traditional setup rather than the one constructed. I am not sure whether the construction of the authors is an appropriate toy model of a real network's operation.
> > * In figure 4, a percentage sign appears in brackets after the $y$-axis name, this would imply that all of the features occupy less than $1$% of the variance.
> > * There does not seem to be a good definition provided for what is an easy or hard feature. This seems to be a classification made by the authors using intuition.
> > * There are also some sentences which remain in the paper which I don't think have appropriate justification. For example, "Thus, the number of possible output values a feature can take seem to strongly affect the variance its representations carry near the output—which seems reasonable in retrospect, given that each possible value that can be output is effectively a separate output logit."
> > * I am not sure how I feel about using the term "total variance" given that separate linear probes are trained on the same activation space and the variance explained is measured using the $R^2$ of each of these separate probes. Might this not fail if the representations are not independent? I am not sure of this point so am open to changing my mind based on rebuttal from the authors.
> >
> > Although I do like the way that the paper has changed, I still not yet at the stage where I will feel comfortable changing to affirmative for "Claims And Evidence".

---

> ### Author Response · Authors · 2024-08-05
> **Second response: some additional corrections, but one paper cannot test everything (1/2)**
>
> We thank the reviewer for their attentive reading and further thoughts. We have fixed the axis issues in the figures and added some clarifying statements as elaborated below. We would like to emphasize that no paper can test—or even discuss—all the potential factors that might possibly affect the results. In this paper (largely the supplement), we have performed ablations on more than 10 distinct factors that could, potentially, affect the MLP results. These include architectural and training details (depth, width, aspect ratio, residual connections, nonlinearity, initialization scale, dropout, optimizer), as well as testing the effect of redundantly encoding the easy feature, testing eight different functions of varying complexity for the hard feature, the interacting effects when four features are learned, etc. In addition, we validate the results across Transformers and CNNs to show that they are not driven purely by simple binary inputs and outputs, and not restricted purely to MLPs. We believe that relatively few interpretability papers (or TMLR publications) have ablated as many parameters to understand how they interact with the main findings, or reproduced their findings across as many distinct settings. We have addressed the specific comments below, and hope that this will satisfy the reviewer.
>
> "Perhaps the authors could include a footnote mirroring the statement in their rebuttal that [...]"
> * We did, in fact, include a statement along these lines, both in the relevant section ("thus it can account for any linear and non-linear patterns of encoding the hard input, *as long as they do not mix in other irrelevant inputs*") and more comprehensively in the limitations ("While these analyses can account for many forms of representation—particularly in §3.3, where the augmented all-input-patterns space can account for any function of the hard inputs alone—they are limited to information that is not linearly available from that augmented space. For example, if the hard features were represented in superposition (cf. Elhage et al., 2022) with other noise inputs, a linear analysis from solely the hard feature space could not account for that.").
>
> "The work only considers binary features. This is okay but I don't think this limitation is discussed enough."
> * This work does *not* only consider binary features, either in the input or the output; it also considers continuous inputs and multinomial outputs (in the vision and/or language tasks). This work *does* focus solely on discrete classification tasks, unlike the continuous tasks the reviewer suggests, but this issue is already discussed in the limitations section: "Moreover, we focused on classification because, again, it made the computational solution more straightforward to analyze. It would be interesting to consider how these biases differ under other objectives—though an increasing amount of work has converged on some kind of (sequential, soft) classification, e.g. language modeling."
>
> "Related to the above, I am concerned about whether the training setup used is really transferable outside of the context of the paper. I could not find in the paper what loss function is used for training the models analysed however this factor could have a large impact on the results."
> * We thank the reviewer for pointing out that the loss was omitted from the training details; we have rectified this. The models were trained with sigmoid cross-entropy loss for the binary classifications, and softmax cross-entropy loss for the multinomial classifications (for the vision and language tasks). These loss functions are among the most common in modern machine learning; for example, as noted in the quote above, the softmax cross-entropy objective is precisely the standard objective used to train transformer language models.
>
> "In figure 4, a percentage sign appears in brackets after the y-axis name, this would imply that all of the features occupy less than 1% of the variance."
> * Indeed, the values were scaled as a proportion from 0-1; we have fixed this issue in this and other figures.
>
> "There does not seem to be a good definition provided for what is an easy or hard feature. This seems to be a classification made by the authors using intuition."
> * We have added a definition of difficulty in terms of the time (i.e., the number of updates) required to learn the task, as well as a note that this is associated with other measures of difficulty, such as the dataset size required to generalize the feature, or the probability that it is computed by a randomly-initialized network.

---

> > ### Author Response · Authors · 2024-08-05
> > **Second response: some additional corrections, but one paper cannot test everything (2/2)**
> >
> > "There are also some sentences which remain in the paper which I don't think have appropriate justification. For example, "Thus, the number of possible output values a feature can take seem to strongly affect the variance its representations carry near the output [...]""
> > * Could the reviewer please clarify in what way this sentence is not justified, e.g. by providing an alternative explanation of the results that is incompatible with it?
> >
> > "I am not sure how I feel about using the term "total variance" given that separate linear probes are trained on the same activation space and the variance explained is measured using the R^2 of each of these separate probes."
> > * By total variance we mean literally the total variance of the neurons across the dataset, i.e. the sum of the squared deviations from their average activity. Total variance does not require regressions to calculate. The method of calculating total variance was already presented in section 2.1, but we have edited the discussion there to further clarify this point.
> >
> > Once again, we thank the reviewer for their attentive comments; we hope that these responses will convince the reviewer that the paper meets the necessary standard of claims and evidence. If the reviewer chooses to recommend our paper for acceptance, we would be happy to correct any further minor omissions or clarifications in the camera ready version.

---

> > > ### Comment · Reviewer_GLvA · 2024-08-06
> > > **Response to comment (2/2)**
> > >
> > > In response to these additional points:
> > > 1. *"Could the reviewer please clarify in what way this sentence is not justified."* In my reading, there isn't sufficient empirical evidence to justify the current language. I would soften the statement to use the word "may" instead of "seem" and remove the word strong. (This is a fairly minor gripe).
> > > 2. *"By total variance we mean literally the total variance..."* I understand the calculation of total variance. My issue is that using this phrase can make it seem as though each feature is taking up an independent percentage of the total variance. However, unless the representations are independent and explain all of the variance, they won't add to the total variance.
> > >
> > > I think after all the changes implemented, I am now happy to change to "Yes" in "Claims and Evidence".

---

> > > > ### Author Response · Authors · 2024-08-07
> > > > **Thank you again**
> > > >
> > > > We thank the reviewer for changing the status of the claims and evidence — and appreciate their thoughtful comments again. We have added some brief comments on some of these points, for example the fact that the variance explained by one feature could, in principle, overlap with that explained by another; however, because we constructed the datasets such that the features are statistically independent, this should not happen in practice, and indeed we find in practice that the variance explained by including the two features in the regression together is not lower than the sum of that explained by the features individually. We've also rephrased the hard feature definition to focus on the perhaps less tautological—and more mathematically precise—definition in terms of the minimal width and depth ReLU network that computes each feature, while mentioning the relationship of this property with learning time and dataset size. Thanks!

---

> > ### Comment · Reviewer_GLvA · 2024-08-06
> > **Response to comment**
> >
> > I think the author's for their detailed response. A few points:
> > 1. When I describe a binary feature I am talking about the individual output dimensions rather than the number of output dimensions. I.e. whether the feature is present or is not. My issue here is that in "real networks" the features represented might not be binary but instead continuous. Again, it's okay to assume that it's binary but without the description of training, I was unsure whether this assumption was in effect or not.
> > 2. Related to this, I think that it's okay to use sigmoid cross-entropy to do classification on binary features. However, I think my point still stands about the generalisation of this to real networks. In a real network, even if the last layer is cross-entropy, the effective loss function on features in an early layer might be very different. It is good that this is brought up in the limitations, but for me, the limitation only made sense once the loss function used was actually reported.
> > 3. If the definition of hard feature is the number of updates required to learn the task, there may be some tautologies in the text. For example, "The easy feature (reading out a single input) is learned quite rapidly, but the hard feature (sum of 4 inputs mod 2) is learned more slowly." (This is a very minor comment).

---

### Author Response · Authors · 2024-07-26
**Response & changes in this revision**

We thank the reviewers for their thoughtful reviews. In response, we have revised the paper as follows:

* Expanded on the discussion of analyses based on all hard input patterns — including the fact that this analysis can account for all linear or non-linear functions of the hard inputs, and thus accomodates many forms of nonlinear representation of the hard feature — but also addressing its limitations.
* Revised the contributions in the introduction into bullet points to highlight takeaways, and revised subsection titles to clarify the main points made by each.
* Added more formal definitions for some concepts (e.g. features) and revised the formal definitions for the feature variance measures in light of them; emphasized the formal definitions for the variance explained etc. in the methods further.
* Added discussion in the related work section of mechanistic interpretability of algorithmic tasks and grokking, and how the current work complements those prior efforts.
* Expanded the discussion of limitations from a paragraph to a subsection, particularly highlighting the limitations of the use of synthetic datasets, and the use of linear analyses, as well as providing more thorough references and suggestions for future work.
* Added references to material in the appendices — like examples of the vision datasets and architectural details — where references were missing from the main text.
* Fixed formatting issues (particularly with figures) that were introduced by some last-minute edits — apologies for those.

We will add further responses to individual reviewers.

---

### Decision · Action_Editor_KGm1 · 2024-09-20

**Recommendation:** Accept as is

**Comment:**

This paper finds that feature representations of DNNs are usually biased by the feature complexity, the order of features in learning, and the distribution of features. These finding are of interests to the community of artificial intelligence. Experiments have well supported these findings. Reviewers agree the acceptance of this paper.

**Audience:**

Findings of this paper would be interested by researchers in artificial intelligence.

**Claims And Evidence:**

Claims made in the submission have been supported by accurate, convincing and clear evidence in experiments.